# A novel mechanosensitive channel controls osmoregulation, differentiation, and infectivity in *Trypanosoma cruzi*

Noopur Dave[1], Ugur Cetiner[2], Daniel Arroyo[1], Joshua Fonbuena[1], Megna Tiwari[1], Patricia Barrera[3], Noelia Lander[4], Andriy Anishkin[2], Sergei Sukharev[2], Veronica Jimenez[1]*

[1]Department of Biological Science, College of Natural Sciences and Mathematics, California State University Fullerton, Fullerton, United States; [2]Department of Biology, University of Maryland, College Park, United States; [3]Departmento de Biología, Facultad de Ciencias Exactas y Naturales, Instituto de Histologia y Embriologia IHEM-CONICET, Facultad de Medicina, Universidad Nacional de Cuyo, Mendoza, Argentina; [4]Department of Biological Sciences, University of Cincinnati, Cincinnati, United States

**Abstract** The causative agent of Chagas disease undergoes drastic morphological and biochemical modifications as it passes between hosts and transitions from extracellular to intracellular stages. The osmotic and mechanical aspects of these cellular transformations are not understood. Here we identify and characterize a novel mechanosensitive channel in *Trypanosoma cruzi* (TcMscS) belonging to the superfamily of small-conductance mechanosensitive channels (MscS). TcMscS is activated by membrane tension and forms a large pore permeable to anions, cations, and small osmolytes. The channel changes its location from the contractile vacuole complex in epimastigotes to the plasma membrane as the parasites develop into intracellular amastigotes. TcMscS knockout parasites show significant fitness defects, including increased cell volume, calcium dysregulation, impaired differentiation, and a dramatic decrease in infectivity. Our work provides mechanistic insights into components supporting pathogen adaptation inside the host, thus opening the exploration of mechanosensation as a prerequisite for protozoan infectivity.

*For correspondence:
vjimenezortiz@fullerton.edu

Competing interests: The authors declare that no competing interests exist.

## Introduction

Mechanosensation is a universal characteristic of all cells, from bacteria to mammals (*Cox et al., 2018*; *Arnadóttir and Chalfie, 2010*). Mechanosensitive channels (MSCs) act as sensors (transducers) and often as effectors of mechanoresponses (*Peyronnet et al., 2014*), and their activation leads to the movement of ions and small osmolytes, which mediates regulatory volume responses and/or triggers downstream signaling cascades (*Martinac et al., 1987*; *Sukharev et al., 1993*; *Martinac, 2011*; *Cox et al., 2013*; *Maksaev and Haswell, 2013*). Thus, mechanosensation allows cells to respond to stress conditions, maintaining a relatively constant volume and macromolecular excluded volume of the cytoplasm. In eukaryotic organisms, mechanosensation is involved in shear stress sensing and stem cell differentiation (*Li et al., 2014*; *Miura et al., 2015*; *Corrigan et al., 2018*), regulating the fate of mesenchymatic tissues such as myoblasts and osteoblasts (*Engler et al., 2006*; *He et al., 2019*). Importantly, mechanical cues regulate the growth of normal and tumoral tissue (*Koser et al., 2016*; *Irvine and Shraiman, 2017*; *Shraiman, 2005*), allowing for the invasion of the basal lamina and increasing the metastasis of tumors (*Barnes et al., 2017*; *Broders-Bondon et al., 2018*). MSCs in prokaryotes have been postulated to participate in quorum sensing, biofilm formation, and regulation of virulence factors (*Siryaporn et al., 2014*;

*O'Loughlin et al., 2013*; *Thomen et al., 2017*; *Prindle et al., 2015*). Protozoan pathogens, like trypanosomes and apicomplexans, are subjected to mechanical forces associated with shear stress in the bloodstream (*Harker et al., 2014*; *Sumpio et al., 2015*), extravasation (*Laperchia et al., 2016*; *Coates et al., 2013*), and invasion of tissues required for establishing persistent infection (*Barragan et al., 2005*; *Barragan and Sibley, 2002*; *Capewell et al., 2016*; *Trindade et al., 2016*; *Mattos et al., 2019*). While it is clear that mechanical cues modulate the parasite's life cycle and behavior, the molecular mechanism underlying sensing and triggering of the responses required for survival is unknown.

*Trypanosoma cruzi*, a flagellated protozoan, is the causative agent of Chagas disease, an endemic pathology in Latin America where millions are infected (*Pérez-Molina and Molina, 2018*). Recent studies indicate global changes in the epidemiology of the disease, resulting in an increased incidence in non-endemic regions of the world (*Requena-Méndez et al., 2015*; *Bern and Montgomery, 2009*). Similar to other protozoan pathogens (*Jimenez, 2014*; *Cowman et al., 2016*; *Silvester et al., 2017*), *T. cruzi* completes its life cycle by alternating between mammalian hosts and insect vectors. The parasite undergoes transitions between intracellular and extracellular forms, with periods of exposure to the environment. Drastic morphological and biochemical changes are required to ensure its survival in different environments (*Jimenez, 2014*). *T. cruzi* epimastigotes replicate in the intestinal lumen of the insect vector where they endure osmotic concentrations of rectal material up to 1000 mOsm/kg (*Kollien and Schaub, 1998*). The infective metacyclic trypomastigotes are released into the environment with the feces of the vector and are able to penetrate the skin layers of a mammalian host, disseminating via blood circulation. The highly motile flagellated trypomastigotes survive shear stress in the bloodstream and then penetrate into host cells where, in the relatively constant cytoplasmic environment, they transform into the round non-motile intracellular amastigotes. The differentiation back to the extracellular infective trypomastigotes requires substantial volume and shape changes, cytoskeletal rearrangement, and elongation of the flagella. Additionally, *T. cruzi* has a strong tropism toward muscle cells, establishing chronic infection in cardiomyocytes and smooth muscle of the gastrointestinal tract (*Ward et al., 2020*). Parasite survival and growth in contractile tissues as well as developmental transitions from intracellular to extracellular forms poses significant mechanical challenges for the parasites (*Blair and Pruitt, 2020*).

Multiple signaling pathways are activated during osmotic responses, invasion, and differentiation (*Rohloff et al., 2004*; *Furuya et al., 2000*; *Moreno et al., 1994*; *Tardieux et al., 1994*), including cAMP and calcium-dependent cascades (*Lander et al., 2021*; *Chiurillo et al., 2017*), but the primary molecular sensors triggering these processes have yet to be identified. *T. cruzi* regulates its volume via the contractile vacuole complex (CVC) (*Rohloff and Docampo, 2008*; *Docampo et al., 2013*), a specialized organelle also present in other protozoans like *Paramecium*, *Dictyostelium*, and *Chlamydomonas*, which experience substantial osmotic gradients across their plasma membranes. The CVC actively collects excessive fluid from the cytoplasm and ejects it through exocytic mechanisms (*Allen and Naitoh, 2002*). In *Chlamydomonas*, yeast, and various higher plants, the mechanisms of volume adjustment, stress relief, biogenesis, and maintenance of intracellular organelles such as plastids are regulated by mechanosensitive channels (*Palmer et al., 2001*; *Lee et al., 2016*; *Peyronnet et al., 2008*; *Haswell and Meyerowitz, 2006*). These channels belonging to the prokaryotic small-conductance mechanosensitive channel (MscS) family act as ubiquitous turgor regulators and osmolyte release valves in bacteria (*Martinac et al., 1987*; *Sukharev et al., 1994*; *Levina et al., 1999*) and archaea (*Kloda and Martinac, 2001*). Currently, there is no information about such tension-stabilizing components in protozoan CVCs.

Here, we report the identification and functional characterization of a novel mechanosensitive channel in *T. cruzi* (TcMscS), which is a member of an MscS-related branch of channels present in trypanosomatids (*Prole and Taylor, 2013*). The sequence analysis predicts a two-transmembrane domain architecture with a unique C-terminal domain. During the parasite's life cycle, the channel is developmentally targeted to CVC in the extracellular motile stages and to the plasma membrane in the intracellular amastigote stage. When expressed in bacterial spheroplasts, the channel gates directly by membrane tension and shows a slight selectivity for anions. Gene targeting by CRISPR/Cas methods generated knockout and knockdown parasites that exhibit impaired growth and inability to robustly regulate cell volume. These parasites also show abnormal calcium regulation and a dramatic decrease in differentiation rate and infectivity, supporting the essential role of mechanosensitive channels in *T. cruzi*. Our results provide evidence that in trypanosomes, TcMscS and its

homologs are involved in the sensing of mechanical forces generated during tissue migration and cell penetration, stabilizing pressure gradients and relieving mechanical membrane stresses during cell volume regulation and developmental transitions.

## Results

### Sequence analysis and structural features of TcMscS predicted by homology model

Analysis of Trypanosoma databases (tritrypdb.org) revealed the presence of a putative MscS-like ion channel (TcCLB.504171.40) in the Esmeraldo-like haplotype of the *T. cruzi* CL Brener strain genome (TcMscS). This channel shares 94.5% identity at the protein level with the non-Esmeraldo-like allele (TcCLB.509795.40) and has homologs in *Trypanosoma brucei* (Tb427.10.9030) and *Leishmania major* (LmjF.36.5770), sharing 66 and 58% identity, respectively (*Figure 1A*). Overall sequence conservation between TcMscS and *Escherichia coli* MscS is low (25.4%), but the percentage of identity increases to 38% in the core comprising transmembrane helices TM2 and TM3. The full alignment of four protozoan and two bacterial sequences is shown in *Figure 1—figure supplement 1*. The ORF of TcMscS is 498 nucleotides, predicting a protein of 165 amino acids (17.8 kDa). Assuming that TcMscS follows the same oligomerization pattern as *E. coli* MscS (*Bass et al., 2002*), it is expected to form a heptamer. The refined consensus homology model based on several structure prediction servers (*Figure 1B*) indicates the presence of two transmembrane domains, with a preceding N-terminal segment (red). The chain continues as TM2 (gold), crosses the membrane, makes a short loop, and returns back as pore-forming TM3 (green). The gate of TcMscS is predicted to be formed by the ring of F64, which is in the precise homologous position with the gate-forming L109 of EcMscS.

The labeling of TcMscS helices in *Figure 1* matches the homologous segments of prokaryotic MscS. Because the TM1 segment in TcMscS is too short to cross the membrane, it is modeled in the conformation attached to the inner surface of the membrane (N1). The beta domain (cyan) is the last segment that aligns with EcMscS. The complete TcMscS homology model, which includes the bundled C-terminal segment, is shown in the predicted closed and open conformations (*Figure 1C*). In addition to the F64 constriction (main gate), the beta domain that follows TM3 is predicted to bear a second hydrophobic constriction (*Figure 1C* top panel, pink residues). This hypothetical C-terminal (cytoplasmic) gate formed by L97 (*Figure 1C*, bottom panel, yellow residues) is narrower than the cage constriction present in *E. coli* MscS. The full model also shows the predicted opening transition that widens both gates. The opening was modeled using the extrapolated motion protocol (EMP) (*Anishkin et al., 2008a*; *Anishkin et al., 2008b*), and the movie of the transition is included in *Video 1*. The overall displacement and tilting of TM3 helices on opening transition is modeled to increase the main gate radius from 3.9 to 6.5 Å accompanied by an in-plane area expansion of the protein by approximately 5.0 nm$^2$. The *de novo* modeled flexible linkers lead to the C-terminal amphipathic helical domains hypothesized to form a coiled-coil bundle (blue), similar to what was observed in the crystal structures of MscL (*Anishkin et al., 2003*), another type of bacterial MS channel. The linkers and bundle might serve as a pre-filter to exclude larger osmolytes from occluding the pore. The details of the modeling, an illustration of the position of the simulated complex in the lipid bilayer, and the structure of the C-terminal bundle can be found in 'Materials and Methods' and in *Figure 1—figure supplement 2*. Overall, the structural model prediction supports the formation of a heptameric channel with a higher degree of conservation in the pore-forming region and some key differences in the C-terminal segment, which are unique to TcMscS.

### Electrophysiological characteristics of TcMscS

To evaluate the functional properties of TcMscS, electrophysiological recordings were performed in *E. coli* giant spheroplasts expressing the channel. Expression was done in the MJF641 (Δ7) strain devoid of seven endogenous MS channel genes (*Edwards et al., 2012*). 12 control patches from untransfected spheroplasts showed no MS channel activity in the entire range of pressure stimuli, whereas ~85% of patches from the expressor strain exhibited 2–11 MS channels per patch. Single-channel currents were measured at low protein expression, with one to three channels per patch. *Figure 2A* shows typical single-channel traces recorded under 110 mmHg pressure steps at +40 and −40 mV pipette voltages. The current amplitude of 12 pA translates to 0.40 ± 0.02 nS conductance

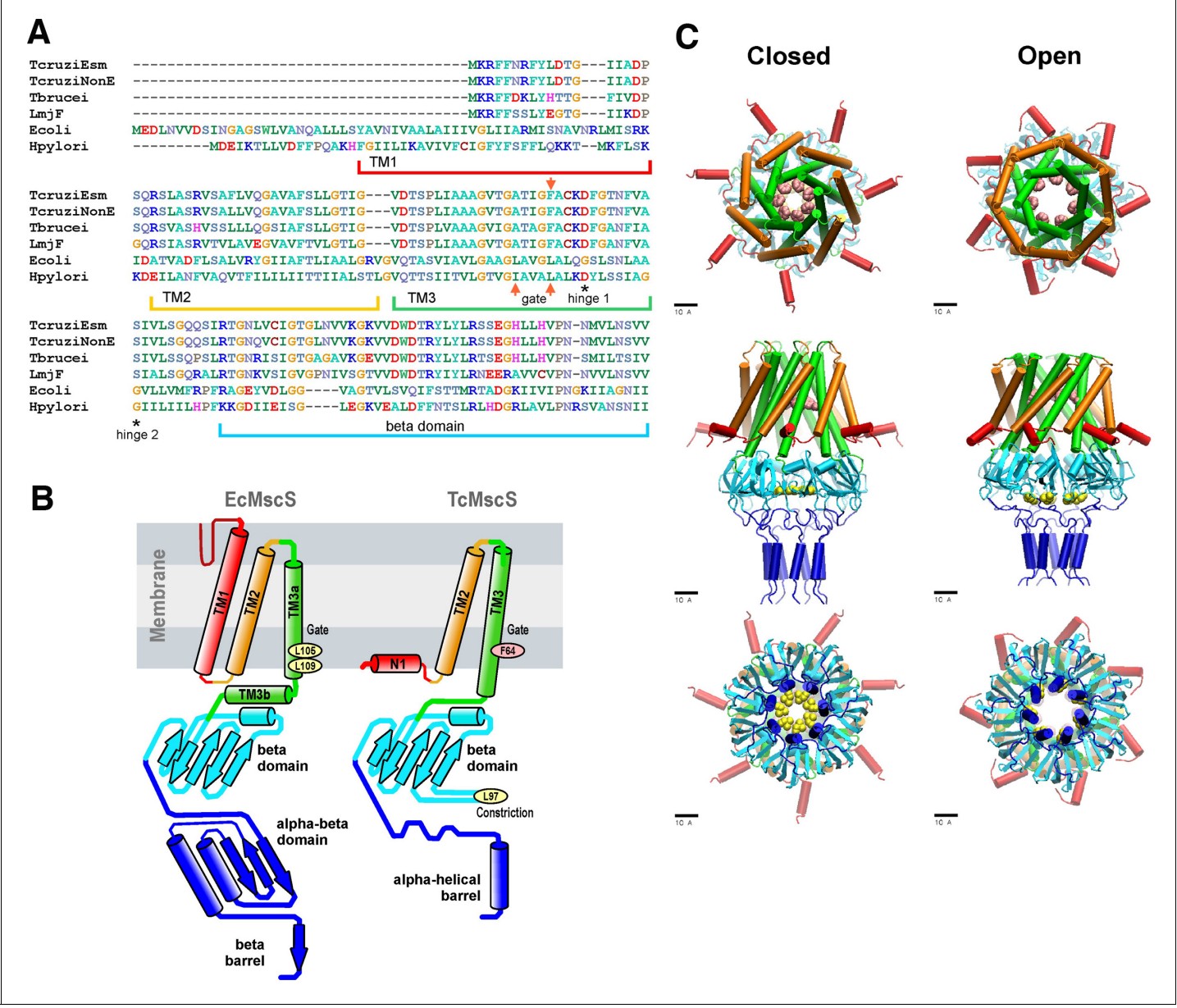

**Figure 1.** Sequence alignment and predictions of TcMscS structure by homology to EcMscS. (**A**) Partial protein sequence alignment of four small-conductance mechanosensitive (MscS)-type channels from *Trypanosoma cruzi*, *T. cruzi* Esmeraldo-like (TcCLB.504171.40) and non-Esmeraldo-like (TcCLB.509795.40) haplotypes of the CL Brener reference strain, *Trypanosoma brucei* (Tb427.10.9030) and *Leishmania major* (LmjF.36.5770), with two bacterial MscS channels from *Escherichia* coli (WP_000389819) and *Helicobacter* pylori (WP_000343449.1). The positions of the transmembrane domains TM1, TM2, and TM3 are underlined, the positions of the putative gate residues are indicated by red arrows, and conserved residues forming the hinges of TM3 are indicated with asterisks. The full alignment can be found in *Figure 1—figure supplement 1*. (**B**) Arrangements of transmembrane helices of EcMscS (PDB ID 2OAU) and in the proposed homology model of TcMscS. The positions of the putative gate and secondary constriction residues for TcMscS (F64 and L97) are indicated. (**C**) Full-homology TcMscS model in the closed and open states. N-terminal domain 1 (red), TM2 (gold), and TM3 (green) are followed by the cytoplasmic cage (cyan) and the C-terminal bundle (blue). The gate residues are indicated in pink and yellow.

The online version of this article includes the following figure supplement(s) for figure 1:

**Figure supplement 1.** Sequence alignment of TcMscS.

**Figure supplement 2.** Molecular dynamic simulations of the TcMscS model.

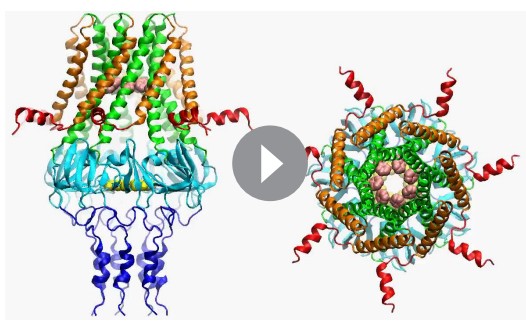

**Video 1.** Opening transition in the modeled TcMscS.
https://elifesciences.org/articles/67449#video1

(n = 11). The I-V curve for TcMscS obtained under symmetric electrolyte conditions (200 mM KCl) is linear (*Figure 2B*, black symbols), in contrast to that for EcMscS, which shows a visible rectification (*Martinac et al., 1987*; *Sukharev, 2002*). After a perfusion of buffer containing 425 mM KCl in the bath (*Figure 2B*, red symbols), the I-V curve changes its slope and the x-intercept reproducibly shifts to $-4.3 \pm 0.4$ mV (n = 4). According to the Goldman equation, under a given ionic gradient, the reversal potential of $-4.3$ mV corresponds to a $P_{Cl^-}/P_{K^+}$ of 1.6, signifying that the channel passes about 3 $Cl^-$ per 2 $K^+$ ions, that is, it shows a slight anionic preference.

To explore whether the channel is permeable to $Ca^{2+}$, we filled the pipette with 100 mM Ca gluconate (supplemented with 20 mM $CaCl_2$ to ensure the silver chloride electrode works) and recorded unitary currents under asymmetric conditions with the standard 200 KCl buffer in the bath. Under a positive pipette voltage (+30 mV), we recorded $12 \pm 1$ pA currents contributed by the inward flux of $Ca^{2+}$ and outward flux of $Cl^-$ (*Figure 2C*). We then thoroughly perfused the bath solution with 100 mM Ca gluconate, making $Ca^{2+}$ the dominant permeable ion at this polarity. The amplitude of current dropped to $0.9 \pm 0.5$ pA (n = 3), translating into a 30 pS conductance (*Figure 2D*). Our previous data *Sukharev, 2002* have indicated that the unitary conductance of EcMscS scales linearly with the bulk conductivity of monovalent salt surrounding the channel. More recently, EcMscS was shown to be highly permeable to $Ca^{2+}$ (*Cox et al., 2013*), and its unitary conductance in 100 mM $CaCl_2$ is about half (0.5 nS) of the conductance in equimolar KCl (1.1 nS). In our experiment with TcMscS, the 400 pS unitary conductance in symmetric 200 mM KCl (24.1 mS/cm) dropped to 30 pS in 120 mM Ca gluconate (4.64 mS/cm). The 5.2-fold drop in bulk conductivity resulted in a thirteenfold decrease in unitary conductance, strongly suggesting that $Ca^{2+}$ is still permeable through TcMscS, although at a lower level compared to monovalents. Interestingly, the TcMscS never opened at negative pipette potentials in the presence of high $Ca^{2+}$.

*Figure 2E* shows two population currents recorded at + 30 or $-30$ mV in the pipette with identical ramp stimuli. The appearance of channels is essentially identical at two opposite voltages. The closing rate appears extremely slow as some channels remain open up to about 10 s after the stimulus ends. More examples of traces with such a 'lingering' channel activity are presented in *Figure 2—figure supplement 1*.

To define the gating characteristics of TcMscS, we analyzed the responses under pressure ramps and normalized the traces to the saturating current to find the open probability and statistically assessed midpoint pressure required for activation (*Figure 2F*). With EcMscS as a reference for the known tension midpoint, we estimated the midpoint tension for TcMscS. EcMscS showed its half-activation pressure at $90 \pm 3$ mmHg (*Figure 2F*, black symbols; n = 54). TcMscS had its pressure midpoint at $145 \pm 20$ mmHg (*Figure 2F*, blue symbols; n = 11). These measurements were done with standard-size pipettes, producing patches of $1.3 \pm 0.2$ µm radius (r) when an activating pressure gradient is applied. Tension in the membrane (T) and the pressure gradient (p) across the patch are linked through the Laplace equation (T = 2 p/r). Given that EcMscS activates in *E. coli* spheroplasts at 7.8 mN/m (*Belyy et al., 2010*), we concluded that the midpoint of TcMscS activation is 12.6 mN/m. Although both channels start opening at approximately the same pressure (tension), the activation curve for TcMscS is shallower. Fitting the initial slopes of logarithmic Po/Pc (open-to-closed probability ratio) plots (panel F, inset) using the two-state Boltzmann equation (*Chiang et al., 2004*) gave estimates of the energies and in-plane area changes associated with gating. Gating of TcMscS was characterized by a 3 $nm^2$ effective expansion of the protein, whereas the slope of EcMscS curve produced a 10.5 $nm^2$ lateral expansion, consistent with models presented in *Figure 1* and the previous work by *Anishkin et al., 2008a*.

These results unequivocally demonstrate that TcMscS is a mechanosensitive channel with functional characteristics comparable to the ones observed in bacterial MscS channels. It opens at sub-lytic tensions and, based on conductance, forms an ~ 12 Å pore, which is essentially non-selective

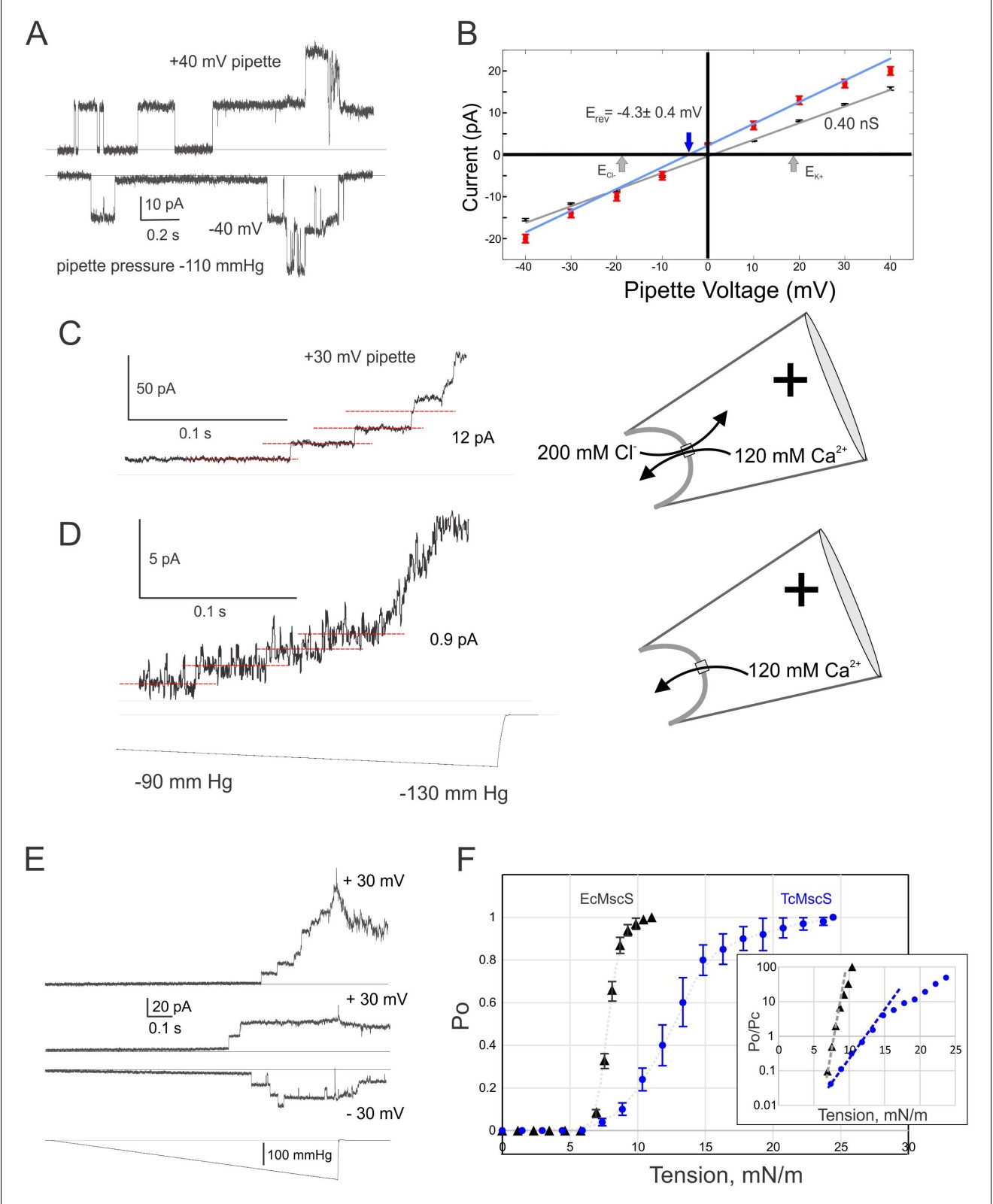

**Figure 2.** Electrophysiological properties of TcMscS. (**A**) Two single-channel traces recorded sequentially under identical steps of a negative pipette pressure of 110 mmHg at pipette voltages +40 mV and –40 mV. (**B**) Current-to-voltage relationships measured in symmetric (200 mM/200 mM KCl, black symbols, n = 11) and asymmetric conditions (425 mM/200 mM KCl, red symbols, n = 4). The theoretical reversal potentials for Cl⁻ and K⁺ (gray arrows) and the experimental reversal potential measured for TcMscS (red arrow) are indicated. (**C**) Unitary conductance of TcMscS recorded with 100

*Figure 2 continued on next page*

*Figure 2 continued*

mM Ca gluconate, 20 mM CaCl$_2$ in the pipette, and 200 mM KCl in the bath, at +30 mV in the pipette. (**D**) The trace recorded from the same patch after perfusion of the bath with 100 mM Ca gluconate. The pressure stimuli for traces in (**C**) and (**D**) were identical ramps shown at the bottom of panel (**D**). In the latter configuration (shown on the right), Ca$^{2+}$ is the dominant permeable ion. (**E**) Examples of traces recorded with identical 1 s linear ramps on three separate patches bearing 11, 2, and 3 active channels at +30 mV or −30 mV pipette potential. (**F**) Multiple ramp responses of TcMscS populations (**E**) were normalized to the current at the saturating pressure and converted to dose-response curves. The aggregated TcMscS curve (blue symbols, n = 11) is compared with EcMscS dose-response curves (black symbols, n = 54). The inset in panel (**F**) shows the same dose-response curves plotted as log (Po/Pc) and fitted with the Boltzmann equation. Po and Pc are the population-normalized open and closed probabilities. The shallower slope of the TcMscS dose-response curve indicates a smaller lateral expansion of the protein complex.

The online version of this article includes the following source data and figure supplement(s) for figure 2:

**Source data 1.** Electrophysiological data analysis.

**Figure supplement 1.** Examples of traces showing slow closure of TcMscS and a 'lingering' channel activity lasting for seconds after the end of the mechanical stimulus.

and capable of passing both anions and cations, including calcium, and possibly organic osmolytes such as betaine, organic acids, and amino acids.

## TcMscS expression and localization

To determine TcMscS localization, we generated polyclonal antibodies against the recombinant protein (as described in 'Materials and methods'). By immunofluorescence analysis (IFA) (*Figure 3A*), we found TcMscS mostly localized in the bladder of the CVC of epimastigotes (*Figure 3A*) and trypomastigotes (*Figure 3B*). In epimastigotes, TcMscS partially colocalizes with VAMP7, another CVC protein (*Ulrich et al., 2011*; *Niyogi et al., 2015*; *Figure 4A*), and with actin (*Figure 4B*). No colocalization with tubulin was observed (*Figure 4C*). In trypomastigotes, TcMscS is found in the CVC bladder and is also distributed along the flagellum and the cell body (*Figure 3B*) but shows no colocalization with the membrane marker SSP-1 (*Figure 4D*). This localization could correspond to intracellular vesicles in traffic to the CVC or to the flagellar attachment zone, which expand along the body of the parasites. In intracellular amastigotes, TcMscS has a peripheral localization and no labeling was detected in the CVC (*Figure 3C*). Colocalization with the amastigote membrane marker SSP-4 shows a clear overlap of the labeling (*Figure 4E*), indicating the translocation of TcMscS to the plasma membrane. The differential localization in the extracellular versus intracellular stages suggests that TcMscS could be playing different physiological roles in response to developmental and/ or environmental cues. This hypothesis is reinforced by our data showing higher expression of TcMscS in epimastigotes compared with amastigotes and trypomastigotes, at the mRNA (*Figure 3D*) and protein levels (*Figure 3E*). Epimastigotes at the stationary phase of growth have a significantly lower amount of transcripts compared with parasites in exponential growth (*Figure 3D*), but no evident difference in the amount of proteins was observed by western blot analysis (data not shown).

## TcMscS gene targeting by CRISPR-Cas9

To establish the physiological role of TcMscS in the parasites, we targeted the gene using the CRISPR-Cas9 system with (*Figure 5*) or without DNA donor for gene replacement (*Figure 5—figure supplement 1*). When we transfected the parasites with the Cas9/pTREX-n vector containing single guide RNA (sgRNA)2 (*Supplementary file 1*-Table 1) targeting *TcMscS* without DNA donor, we observed a single-nucleotide deletion (*del*177G), resulting in a frame shift and a premature stop codon (*Figure 5—figure supplement 1A*, top panel). These results were confirmed by genomic DNA sequencing in multiple independent samples. The truncated protein at amino acid 79 is not expected to be functional (*Figure 5—figure supplement 1A*, bottom panel) since the C-terminal end is absent. In these parasites, the transcript for TcMscS was still detected, although at a significantly lower level compared with control parasites only expressing Cas9 and a scrambled sgRNA (*Figure 5—figure supplement 1C*). Based on the level of mRNA and the detection of residual protein by western blot analysis (*Figure 5—figure supplement 2, A and B*), it is possible that we are only targeting one of the alleles of the gene and we considered this strain a knockdown (TcMscS-KD). Importantly, the growth of TcMscS-KD epimastigotes is significantly reduced (*Figure 5—figure supplement 1B*, blue line) when compared to wild-type Y strain (WT) (*Figure 5—*

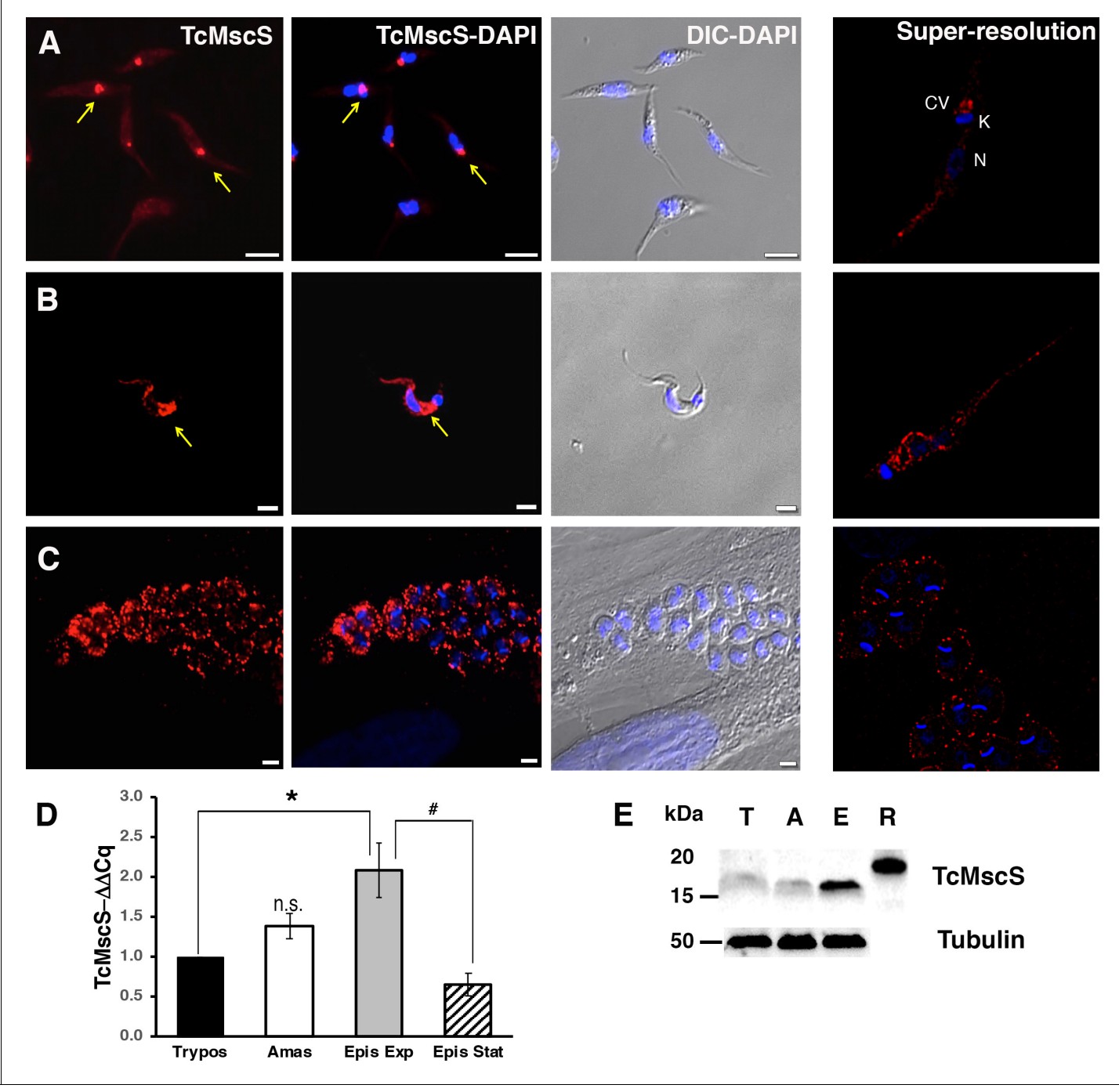

**Figure 3.** Localization and expression of TcMscS in *T.cruzi*. Immunolocalization of TcMscS (red) with specific polyclonal antibodies in epimastigotes (A), trypomastigotes (B), and intracellular amastigotes (C). The right panels correspond to images of the same life stages obtained by super-resolution microscopy. Cellular structures are indicated as CV (contractile vacuole), K (kinetoplast), and N (nucleus). DNA was 4′,6-diamidino-2-phenylindole (DAPI) stained. Bar sizes: (A) 5 µm; (B) and (C) 5 µm. (D) Expression of TcMscS in different life stages of the parasite, quantified by quantitative reverse transcription PCR (RT-qPCR). Trypomastigotes (Tryp) and amastigotes (Amast) were obtained from HEK-293 cells. Epimastigotes were analyzed at 4 days (Epis Exp) and 10 days of growth (Epis Stat). The values are indicated as ΔΔCq with respect to the expression in trypomastigotes and normalized against GAPDH as a housekeeping gene. The values are mean ± SEM of three independent experiments in triplicate (*p = 0.006, #p = 0.009). (E) Western blot analysis of TcMscS in trypomastigotes (T), amastigotes (A), and epimastigotes (E). Purified recombinant protein (R) was used as the positive control. The whole-cell lysates were probed with polyclonal antibodies against TcMscS and monoclonal anti-tubulin was used as the loading control.

The online version of this article includes the following source data for figure 3:

**Source data 1.** Data source for *Figure 3* provided as *Figure 3—source data 1*.

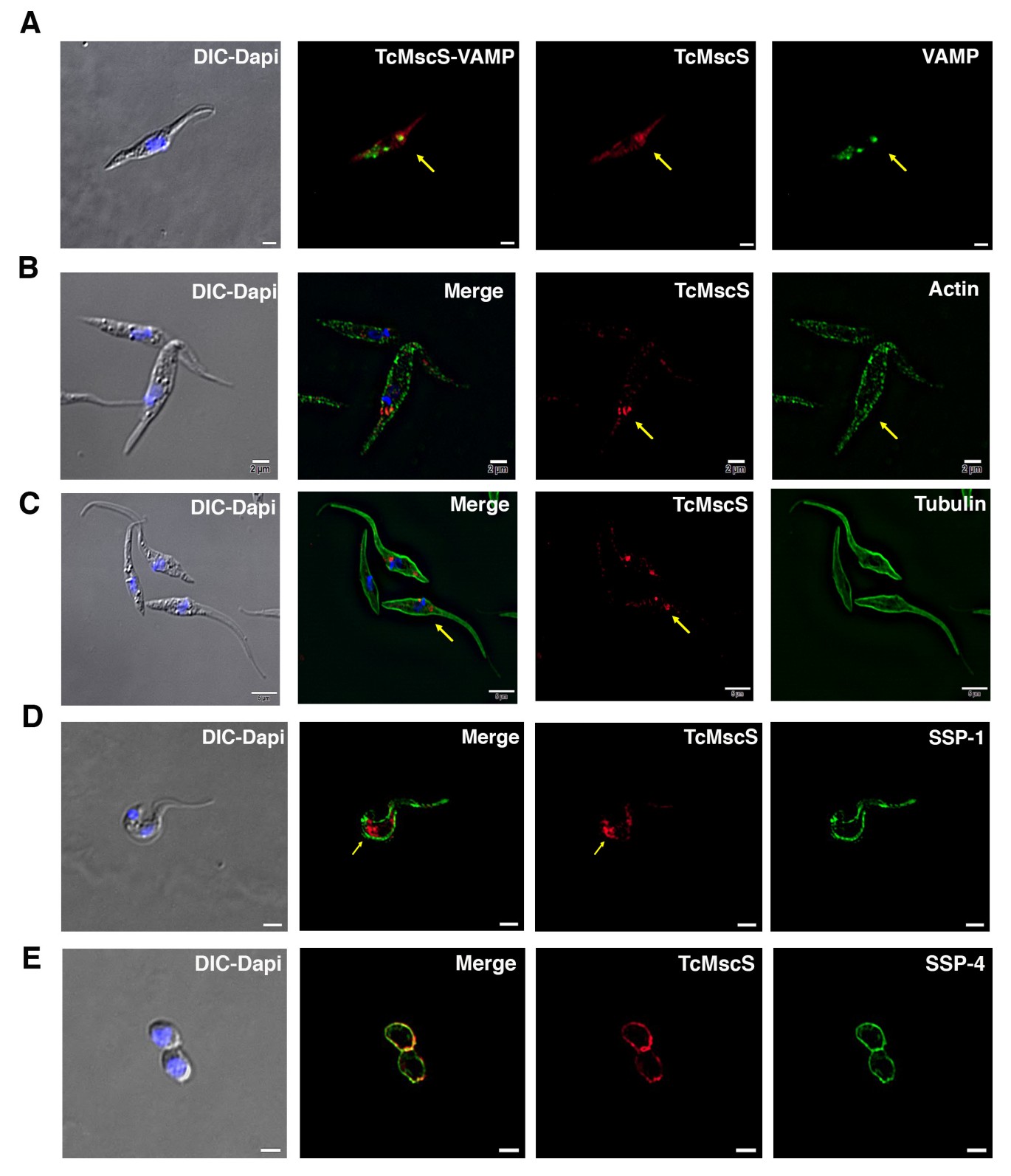

**Figure 4.** TcMscS colocalization with organelle markers. (**A**) Immunofluorescence analysis of TcMscS (red) localization in epimastigotes overexpressing TcVAMP-GFP (green). (**B**) Labeling of TcMscS (red) in wild-type Y strain (WT) epimastigotes partially colocalizes with anti-actin antibodies (green). (**C**) Localization of TcMscS (red) and tubulin (green) in Y strain epimastigotes. No significant colocalization was observed. (**D**) Immunolocalization of TcMscS (red) and membrane marker SSP-1 (green) in Y strain tissue-derived trypomastigotes. (**E**) Colocalization of TcMscS (red) and membrane marker SSP-4

*Figure 4 continued on next page*

*Figure 4 continued*

(green) in extracellular amastigotes obtained by host cell lysis. Nuclei and kinetoplasts were 4',6-diamidino-2-phenylindole (DAPI) stained. Arrows indicate the position of the contractile vacuole. Bar size: 5 µm.

---

*figure supplement 1B*, black line) and Cas9-scr controls (*Figure 5—figure supplement 1B*, red line). The parasites showed an abnormal morphology, with rounded bodies and formation of rosettes (*Figure 5—figure supplement 1D*, bottom panel).

To completely abolish the expression of TcMscS, we co-transfected the parasites with sgRNA2-Cas9/pTREX-n vector plus a donor DNA cassette for homology-directed repair (*Figure 5A*). Genomic DNA analysis by PCR shows the absence of the *TcMscS* gene and the correct insertion of the blasticidin resistance cassette (*Figure 5B*). No transcript was detected in epimastigotes (*Figure 5D*) or trypomastigotes (*Figure 5—figure supplement 2, C*). These results were confirmed by western blot analysis (*Figure 5—figure supplement 2, A*) and immunofluorescence (*Figure 5E*), demonstrating the successful knockout of *TcMscS* (TcMscS-KO).

As expected, the growth rate of TcMscS-KO epimastigotes was significantly reduced (*Figure 5C*, green line) and was lower than that in TcMscS-KD parasites (*Figure 5C*, blue line). The decrease in growth is accompanied by a marked reduction in motility (*Videos 2* and *3*), showing that the ablation of the channel has a negative effect on parasite fitness. To verify the specificity of the phenotype, we complemented the KO strains with the overexpression vector pTREX carrying a copy of TcMscS C-terminally tagged with myc. We successfully expressed the tagged construct (*Figure 5—figure supplement 3A*, C1 and C2 correspond to two clones of the complemented strains), but as expected, based on our previous data, the overexpression has a detrimental effect on the cells. This is in agreement with previous reports of porin and porin-like overexpression toxicity (*Aistleitner et al., 2013*; *Ghosh et al., 1998*). The cell growth was not recovered (*Figure 5—figure supplement 3B*) and the epimastigotes showed an abnormal morphology (*Figure 5—figure supplement 3C*). We then complemented the TcMscS-KO strain with a copy of the *T. brucei* ortholog TbMscS (*Figure 5—figure supplement 3A and B*, C3), and we were able to partially revert the cell growth phenotype, indicating some functional conservation of the channel in other trypanosomatids.

## TcMscS regulates parasites' osmotic stress responses

To evaluate the role of TcMscS in cell volume regulation, we exposed the parasites to osmotic changes and followed the variations in absorbance over time as an indicator of variations in cell volume. Under hypoosmotic conditions of 115 mOsm/l (*Figure 6A*), TcMscS-KD and -KO epimastigotes experienced a maximum change in volume of about 38% compared with 23–25% in the controls (*Supplementary file 2*-Table 2, peak, and *Figure 6B*). When we analyzed the rate of recovery between 200 and 400 s, no significant difference was observed in the slope values, except for TcMscS-KO, which had a slightly higher value. Nevertheless, KD and KO parasites do not regain their normal volume after 600 s, showing a remaining volume change of about 20% (*Supplementary file 2*-Table 2, final volume, *Figure 6C*).

These results indicate that TcMscS, similar to other MscS channels, activates in response to increased membrane tension, which results in redistribution of osmolytes, and thus prevents excessive cell swelling. However, TcMscS-KD and -KO parasites under hypoosmotic stress do not lose control of their volume completely, suggesting that TcMscS must act in conjunction with other mechanisms to orchestrate effective regulatory volume decrease responses.

Upon hyperosmotic stress of 650 mOsm/l (*Figure 6D*), a similar phenotype was observed, with TcMscS-KD and TcMscS-KO parasites reducing their volumes by about 20%, compared with 12–15% in the controls (*Supplementary file 3*-Table 3, peak; *Figure 6E*). As previously described (*Li et al., 2011*), epimastigotes under hyperosmotic conditions do not regain normal volume, and parasites lacking TcMscS remained significantly more shrunken compared to controls (*Figure 6F*), showing that TcMscS is required for regaining normal cell volume following hyperosmotic stress. We evaluated whether TcMscS localization changes upon osmotic stress and observed no differences in parasites under iso-, hypo-, or hyperosmotic conditions at different time points (*Figure 6—figure supplement 1*). It should be noted that TcMscS-KD and -KO epimastigotes under isosmotic conditions show a cell volume about 36–38% higher than normal when compared with WT and Cas9

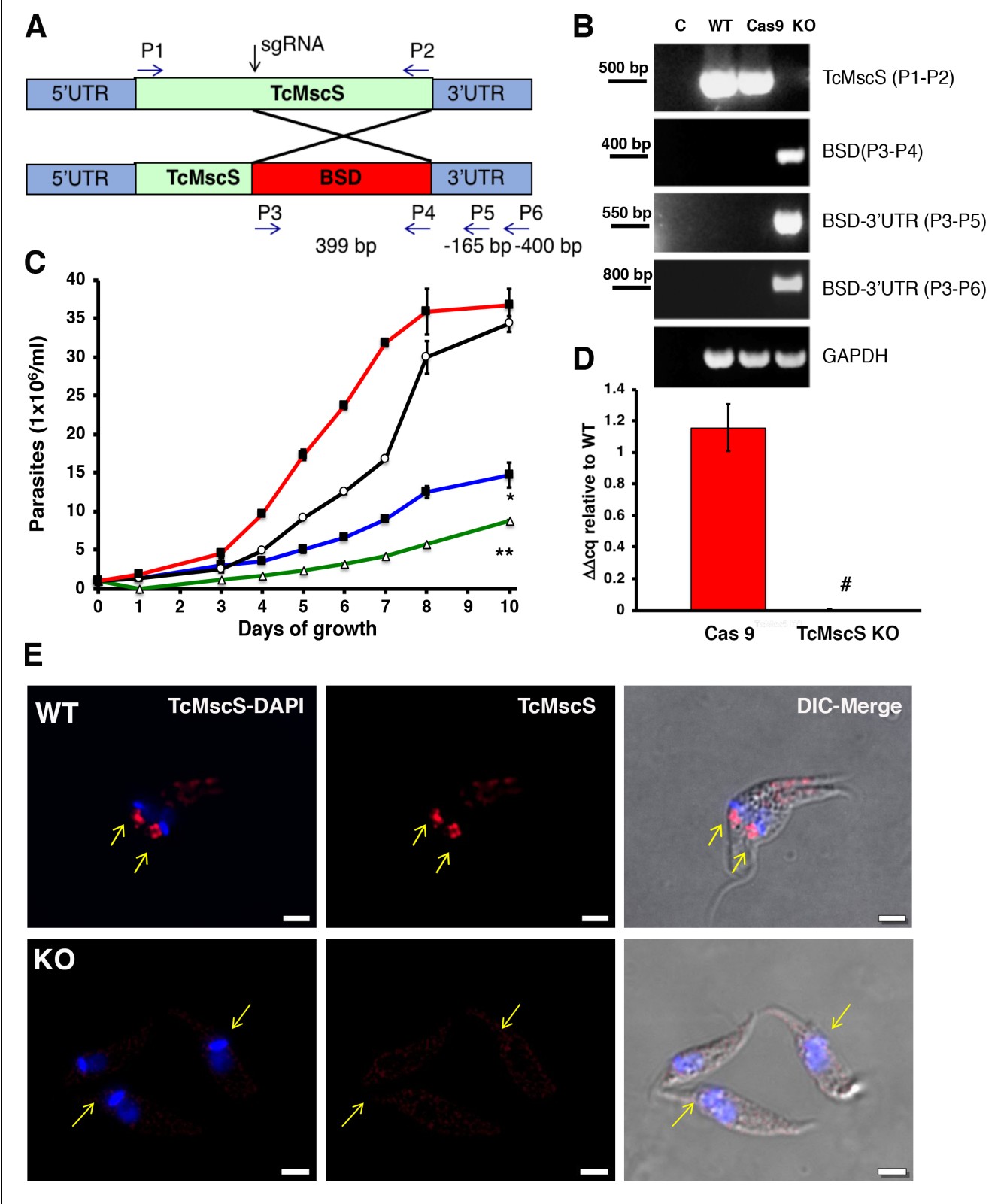

**Figure 5.** Phenotypic characterization of TcMscS-KO. (**A**) Schematic representation of *TcMcS* gene targeting by CRISPR/Cas9. A double-strand DNA cut was induced at position +185 and repaired by homologous recombination with a construct containing the *Bsd* gene. The primers used for screening are listed in *Supplementary file 1*-Table 1. (**B**) Genomic DNA screening of TcMscS-KO epimastigotes by PCR. Negative controls (**C**) without DNA are in lane 1 for all reactions. The ORF for *TcMscS* was amplified in wild-type Y strain (WT) and Cas9 controls but absent in TcMscS-KO parasites. The correct
*Figure 5 continued on next page*

*Figure 5 continued*

insertion of the blasticidin cassette (BSD) was verified by amplification with primers annealing in ORF and the 3'UTR. GAPDH was used as a housekeeping gene for the loading control. (C) Growth curve of epimastigotes WT (black line), Cas9 (red line), TcMscS-KD (blue line), and TcMscS-KO (green line). All the experiments were started at 1 x 10$^6$ cells/ml and counted daily. The values are mean ± SEM of three independent experiments in triplicate (*, **p<0.01). (D) Quantitative reverse transcription PCR (RT-qPCR) quantifying the expression of TcMscS in epimastigotes control (Cas9) vs TcMscS-KO. The values are indicated as ΔΔCq with respect to the expression in WT epimastigotes and normalized against GAPDH as the housekeeping gene. All the samples were collected from parasites at 4 days of growth. The values are mean ± SEM of three independent experiments in triplicate (#p = 0.0001). (E) Immunofluorescence analysis of TcMscS (red) in epimastigotes Y strain (WT), TcMscS-KO. Nuclei and kinetoplasts were 4',6-diamidino-2-phenylindole (DAPI) stained. The position of the contractile vacuole is indicated with yellow arrows. Bar size = 2 μm.

The online version of this article includes the following source data and figure supplement(s) for figure 5:

**Source data 1.** Phenotype analysis of TcMscS KO epimastigotes.
**Figure supplement 1.** TcMscS targeting by CRISPR/Cas9 produced a knockdown effect.
**Figure supplement 2.** TcMscS expression levels in epimastigotes and trypomastigotes.
**Figure supplement 2—source data 1.** Quantification of TcMscS expression.
**Figure supplement 3.** Complementation of TcMscS-KO strains.
**Figure supplement 3—source data 1.** Complementation of TcMscS KO strains.

controls (*Figure 6—figure supplement 2*). This significant difference underscores the role of TcMscS in the regulation of cell volume under normotonic conditions.

## TcMscS-KO causes cytosolic calcium dysregulation

TcMscS localization in the CVC and our electrophysiology data showing the permeation of calcium through the channel lead us to evaluate the cytosolic calcium level in the KO parasites. WT epimastigotes loaded with Fura 2-AM showed a baseline cytosolic Ca$^{2+}$ of 102 ± 4 nM (n = 9) and a robust increase upon addition of 1.8 mM extracellular CaCl$_2$ (*Figure 7A and B*, black line) reaching 208 ± 8 nM 100 s post-stimulation. In contrast, TcMscS-KO had a lower baseline Ca$^{2+}$ of 81 ± 4 nM (n = 8) and a lower increase post-addition (*Figure 7A and B*, green line). The increase in cytosolic-free Ca$^{2+}$ can be caused by release of the ion from intracellular stores and/or by uptake from the extracellular media. To evaluate whether the impairment of the KO was due to a reduced uptake, we treated the cells with Bay K8644, a known activator of voltage-gated calcium channels (VGCC). These channels have been postulated as the main permeation pathway for extracellular Ca$^{2+}$ in *T. cruzi* (*Rodriguez-Duran et al., 2019*), but other reports have indicated its presence in the CVC (*Ulrich et al., 2011*). When we stimulate the cells in the presence of Bay K (*Figure 7C and D*), WT parasites show a dramatic increase in cytosolic Ca$^{2+}$, tenfold higher than in non-pretreated cells (*Figure 7A and C*, black lines). TcMscS-KO exhibited an increase in cytosolic Ca$^{2+}$ compared with non-Bay K conditions (*Figure 7A and C*, green lines), but the response was significantly lower than in WT (*Figure 7D*, post-Bay K). Collectively, these results indicate the role of TcMscS in regulating the cytosolic Ca$^{2+}$ dynamics on the parasites, probably by controlling intracellular stores such as the CVC. The lack of adequate Ca$^{2+}$ homeostasis can severely impact cell motility and infectivity.

## TcMscS-KO impairs parasite differentiation and infectivity

To elucidate the role of TcMscS in *T. cruzi*'s infectivity, we first evaluated the localization of the channel in metacyclic trypomastigotes (*Figure 8A*). Similar to tissue-derived trypomastigotes, TcMscS is localized in the CVC bladder of the metacyclic forms, but also along the cell body, on the flagellar attachment zone region. The differentiation to metacyclic trypomastigotes *in vitro* is severely impaired in TcMscS-KO parasites, producing only 0.38% vs 9.1% obtained from WT epimastigotes after 72 hr of incubation in triatomine artificial urine (TAU)

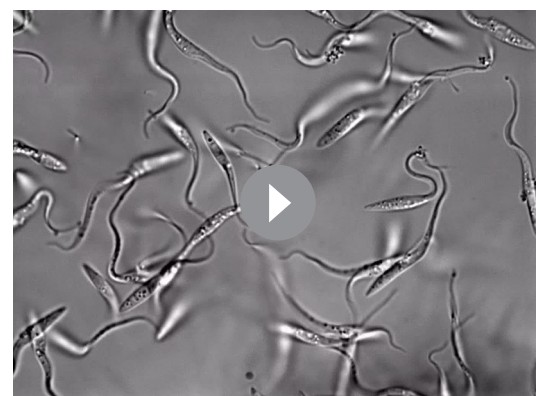

**Video 2.** Motility of Cas9 epimastigotes.
https://elifesciences.org/articles/67449#video2

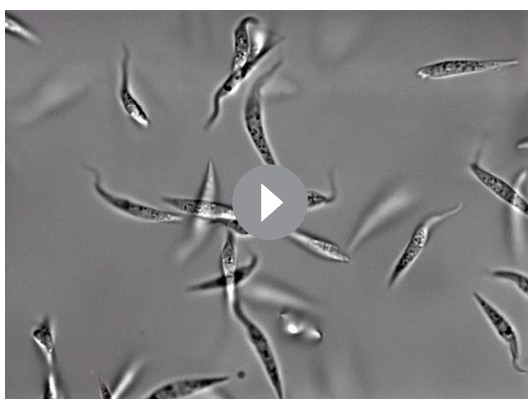

**Video 3.** Motility of TcMscS-KO epimastigotes.
https://elifesciences.org/articles/67449#video3

media (*Figure 8B*; *Contreras et al., 1985*). We were able to infect mammalian cells with TcMscS-KO metacyclic trypomastigotes and obtained tissue culture-derived trypomastigotes. The infective capacity of these parasites was not affected at early time points post-infection (*Supplementary file 4*) but did show a significant decrease in the production of intracellular amastigotes at 48 hr post-infection (*Figure 8C and E*; *Supplementary file 4*). These results confirm a defect in the capacity of the TcMscS-KO to replicate intracellularly. Quantification of extracellular trypomastigotes collected from the tissue culture supernatant shows a significant decrease in the number of TcMscS-KO parasites able to successfully differentiate into bloodstream trypomastigotes and egress from the cells (*Figure 8D*). At 6 days post-infection, TcMscS-KO produced a total of $5.2 \times 10^6$ extracellular parasites, almost one order of magnitude lower than the amount produced by the Cas9-scr strain and WT parasites (*Supplementary file 5*-Table 5).

In summary, the ablation of *TcMscS* results in a loss of parasite fitness, impairing osmotic regulation, motility, calcium homeostasis, differentiation, and cell infection capacity. The multiple phenotypic defects observed in TcMscS-KO parasites indicate that the physiological roles of this mechanosensitive channel have expanded beyond the primary osmotic compensation reported for its bacterial ancestor EcMscS.

## Discussion

Propagation of *T. cruzi* involves alternating between mammalian and insect hosts, a life-history strategy that imposes significant mechanical demands on the parasite. This complex life cycle requires protective mechanisms and a strong differentiation potential that drives massive cell remodeling. The presence of TcMscS homologs in *Toxoplasma gondii*, *Plasmodium falciparum*, and *Entamoeba histolytica* suggests a conserved function for these channels in other protozoan parasites (*Prole and Taylor, 2013*). Understanding their role as molecular transducers as well as likely effectors of mechanical cues throughout the parasite's life cycle can provide new information about the role of physical and osmotic forces in the mechanisms regulating infectivity.

In this work, we identify and present a functional characterization of the first MscS-like mechanosensitive channel in protozoan pathogens. TcMscS, a novel mechanosensitive channel found in *T. cruzi*, is required for growth, calcium homeostasis, differentiation, and infectivity. The channel plays special roles in balancing osmotic and/or contractile forces in a stage- and location-specific manner. TcMscS belongs to the MscS superfamily and is structurally and functionally similar to the well-characterized EcMscS (*Sukharev, 2002*). MscS-like channels are a highly diverse group of proteins *Malcolm and Maurer, 2012* found in archaea (*Kloda and Martinac, 2001*), fungi (*Kumamoto, 2008*; *Nakayama et al., 2012*; *Nakayama et al., 2014*), protozoans (*Tominaga and Naitoh, 1994*; *Yoshimura, 1998*; *Nakayama et al., 2007*), and plants (*Lee et al., 2016*; *Peyronnet et al., 2008*; *Maksaev and Haswell, 2012*; *Wilson et al., 2011*), but absent in metazoans. The evolutionary origin of MscS-like channels can be traced to a common ancestor that was maintained across lineages as a protective mechanism against osmotic stress (*Kloda and Martinac, 2002*). Beyond their primary role in volume regulation, these channels are also implicated in amino acid transport (*Nakayama et al., 2018*; *Becker et al., 1828*), calcium homeostasis (*Nakayama et al., 2012*; *Nakayama et al., 2014*; *Nakayama and Iida, 2014*), and organelle biogenesis and maintenance (*Lee et al., 2016*; *Peyronnet et al., 2008*; *Haswell and Meyerowitz, 2006*; *Maksaev and Haswell, 2012*; *Wilson et al., 2014*; *Wilson and Haswell, 2012*), highlighting the diversification of functions for this protein family over its evolutionary history (*Malcolm and Maurer, 2012*; *Cox et al., 2015*).

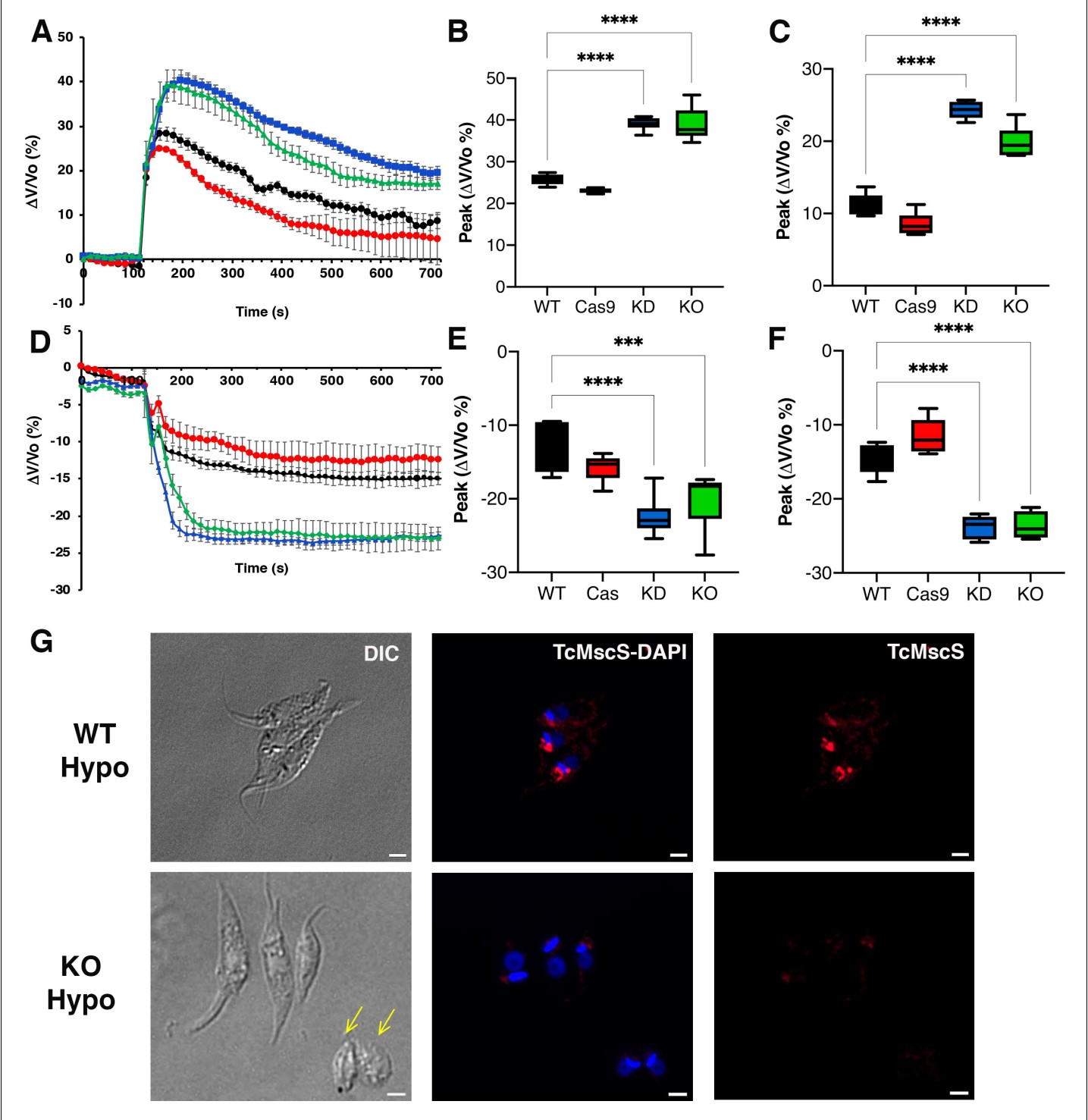

**Figure 6.** Osmotic stress responses in TcMscS mutants. (**A**) Regulatory volume decreases in epimastigotes. Cells suspended in isosmotic buffer recorded for 120 s and diluted to a final osmolarity of 115 mOsm/l under constant ionic conditions. Relative changes in cell volume were monitored by determining absorbance at 550 nm over time, in wild-type Y strain (WT) (black), Cas9 (red), TcMscS-KD (blue), and TcMscS-KO (green). The absorbance values were normalized by the initial volume under isosmotic conditions and expressed as percentage of volume change. (**B**) Analysis of the maximum volume change under hypoosmotic conditions. The peak area was calculated between 150 and 250 s for all the experiments. (**C**) Final volume recovery calculated between 600 and 700 s. (**D**) Volume decrease in epimastigotes after a hyperosmotic shock. After 3 min under isosmotic conditions, cells were placed under hyperosmotic stress at 650 mOsm/l. Volume changes were monitored and calculated as explained in (**A**). Peak analysis (**E**) and final volumes were calculated at the same times as that in (**B**) and (**C**). The values for (**A–F**) are mean ± SEM of six independent experiments in triplicate

*Figure 6 continued on next page*

*Figure 6 continued*

(p<0.001). Statistical analysis details are listed in **Supplementary files 2** and **3**. (G) Representative immunofluorescence images of WT and TcMscS-KO epimastigotes at 2 min post hypoosmotic stress. TcMscS (red) was detected with specific polyclonal antibodies against the channel. Nuclei and kinetoplasts were 4′,6-diamidino-2-phenylindole (DAPI) stained. Yellow arrows indicate the presence of cells with an abnormal morphology, observed in the TcMscS-KO strain. Bar size = 2 μm.

The online version of this article includes the following source data and figure supplement(s) for figure 6:

**Source data 1.** Analysis of osmotic stress responses.

**Figure supplement 1.** Localization of TcMscS in epimastigotes under osmotic stress conditions.

**Figure supplement 2.** Volume of TcMscS mutants under isosmotic conditions.

**Figure supplement 2—source data 1.** Cell volume analysis under normotonic conditions.

Expression in Δ7 *E. coli* spheroplasts provided a system with a silent mechanosensitive background in which to study activation by tension and single-channel properties of TcMscS. The linear non-rectifying conductance of the channel is approximately half of the conductance of EcMscS (**Sukharev, 2002**). Similar to other MscS-like channels, TcMscS has a slight selectivity toward anions, but also shows measurable permeability to calcium.

The size of the pore appears to be sufficient for permeation of small organic osmolytes. In *T. cruzi*, regulatory volume decrease is driven primarily by efflux of amino acids and requires fluctuations in intracellular calcium (**Rohloff et al., 2003**). Given the localization in the CVC and the non-selective nature of TcMscS, this channel could be a permeation pathway for amino acids and calcium required for cell volume regulation. An activation midpoint near 12 mN/m determined in bacterial spheroplasts suggests that the channel becomes fully active at tensions approaching the lytic limit, and thus it fulfills the role of a stress-relieving emergency valve, permitting the equilibration of osmolytes. TcMscS activities recorded in a heterologous expression system strongly suggest that the channel is gated directly by tension in the lipid bilayer without the need for cytoskeleton or other *T. cruzi*-specific components. The ramp responses of TcMscS populations indicate a slow closing rate that allows the channel to remain open for a longer period, thus relieving the hypoosmotic stress more completely.

The channel has a novel 2TM topology (**Figure 1**) and shows substantial homology with *E. coli* MscS only in the TM3 and beta-domain regions. Common to other tension-activated channels (**Blount et al., 1996**), TcMscS is predicted to have a short amphipathic N-terminal helix attached to the inner leaflet of the membrane. TcMscS has a completely unique C-terminal domain predicted to fulfill the function of a pre-filter at the cytoplasmic entrance.

In most cell types, MscS channels are localized in the plasma membrane, but they are also found in internal organelles including the endoplasmic reticulum (**Nakayama et al., 2012**), contractile vacuole (**Palmer et al., 2001**; **Fujiu et al., 2011**; **Plattner, 2013**; **Denis et al., 2002**), and plastids (**Haswell and Meyerowitz, 2006**; **Nakayama et al., 2007**). In *T. cruzi*, TcMscS shows a differential localization in the extracellular versus intracellular stages. In epimastigotes, the channel has a defined intracellular localization in the bladder of the contractile vacuole and partially colocalizes with VAMP-7 and actin. There are no previous reports of actin associated to the bladder of the CVC, but we have shown the expression of myosin heavy chains V1 and V2 (**Ulrich et al., 2011**), suggesting the presence of a contractile machinery that might contribute to the periodic discharge of the organelle, similar to what has been described in *Dictyostelium discoideum and Paramecium* (**Plattner, 2013**; **Tominaga et al., 1998**). In tissue-derived bloodstream and in vitro-differentiated metacyclic trypomastigotes, TcMscS localization is expanded beyond the CVC to include the cell body and part of the flagellum. This suggests that the channel could be sensing changes in tension during flagellar beating. The patched distribution adjacent to the flagellar attachment and along the cell body could be due to accumulation of the channels in vesicles, as it has been reported for other types of channels such as AQP2 and Kv7.1 (**Wang et al., 2020**; **Andersen et al., 2015**).

The role of CVC as a key regulator of the cell volume in *T. cruzi* has been extensively characterized (**Rohloff and Docampo, 2008**), and the presence of a mechanosensitive channel in CVC points to necessity of curbing excessive tensions to protect the organelle from osmotic lysis. Proteomic and functional analysis has shown the presence of numerous proteins involved in volume compensation (**Ulrich et al., 2011**), but no sensor of the osmotic state has been identified. We propose that TcMscS is activated upon excessive filling of the bladder, similar to what has been shown in other

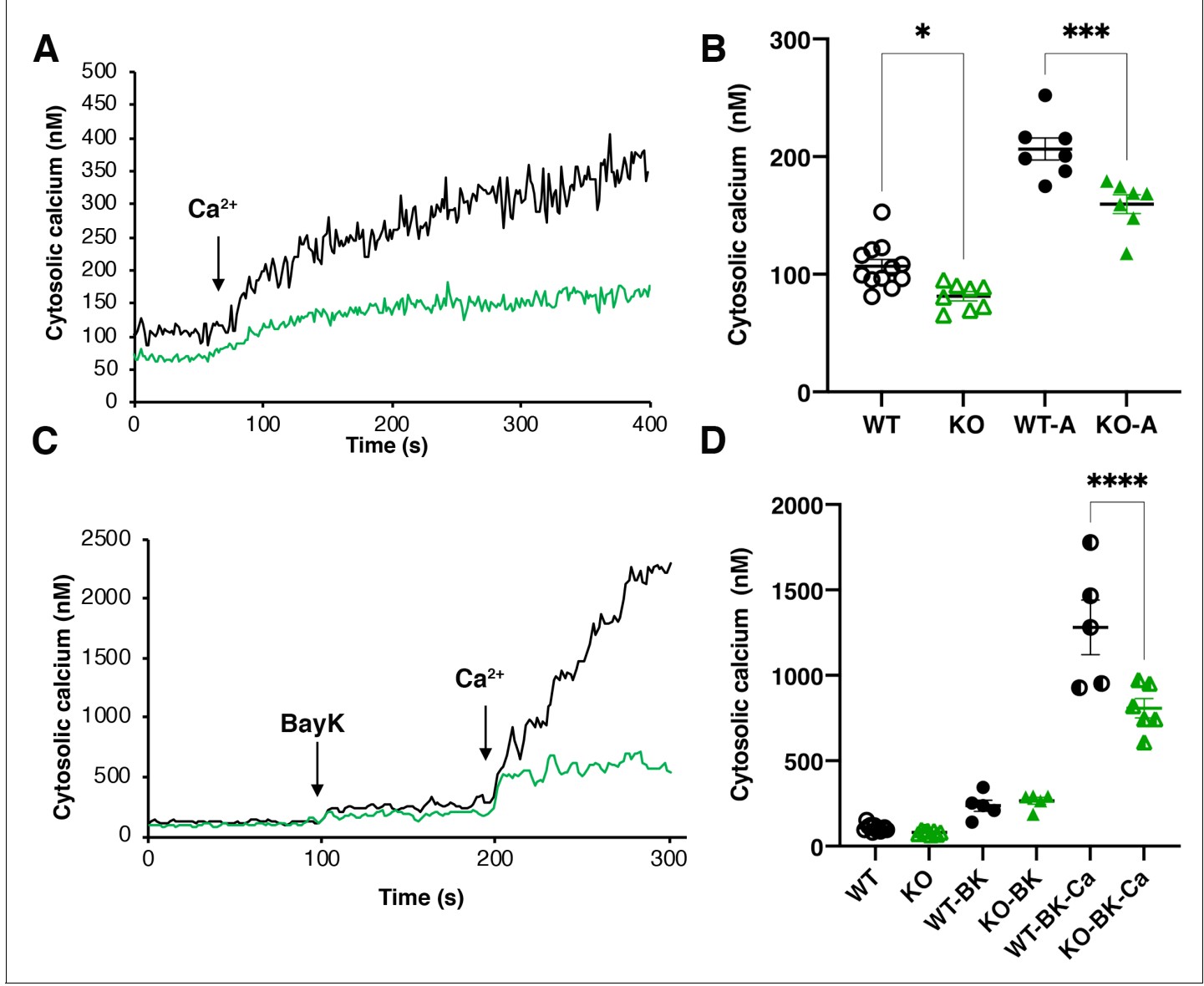

**Figure 7.** Intracellular calcium measurements. (A) and (C) are representative traces of intracellular calcium level in wild-type Y strain (WT) (black line) or TcMscS-KO (green line) epimastigotes loaded with Fura 2-AM at baseline or after the addition of 1.8 mM $CaCl_2$ and 10 µM of Bay K8644 (BK). (B) and (D) show the quantification of baseline $Ca^{2+}$ concentration in the first 75 s WT (black) and TcMscS-KO (green) and in the first 100 s after addition of the stimuli for multiple experiments. In (B), WT-A and KO-A indicate the values after calcium addition. In (D), quantifications were done after addition of Bay K8644 (BK and Bay K plus calcium (BK-Ca)). The values are mean ± SE of n = 7 (B) and n = 5 (D) independent experiments (*p<0.01, ***p<0.001, ****p< 0.0001).

The online version of this article includes the following source data for figure 7:

**Source data 1.** Intracellular calcium measurements data analysis.

protists (*Docampo et al., 2013*; *Tominaga et al., 1998*; *Tani et al., 2000*). If the normal exocytotic release of the bladder content fails to relieve pressure, the channel activation may release the content back to the cytoplasm to preserve the integrity of the CVC.

The peripheral localization of TcMscS in amastigotes deserves further consideration. Intracellular stages develop free in the cytosol under osmotically stable conditions, establishing persistent infection predominantly in muscle cells where contractility generates periodic stress (*Blair and Pruitt, 2020*). It is known that *T. cruzi* infection causes extensive cytoskeletal remodeling and the amastigotes are not protected by a parasitophorous vacuole; thus, they can be mechanically perturbed

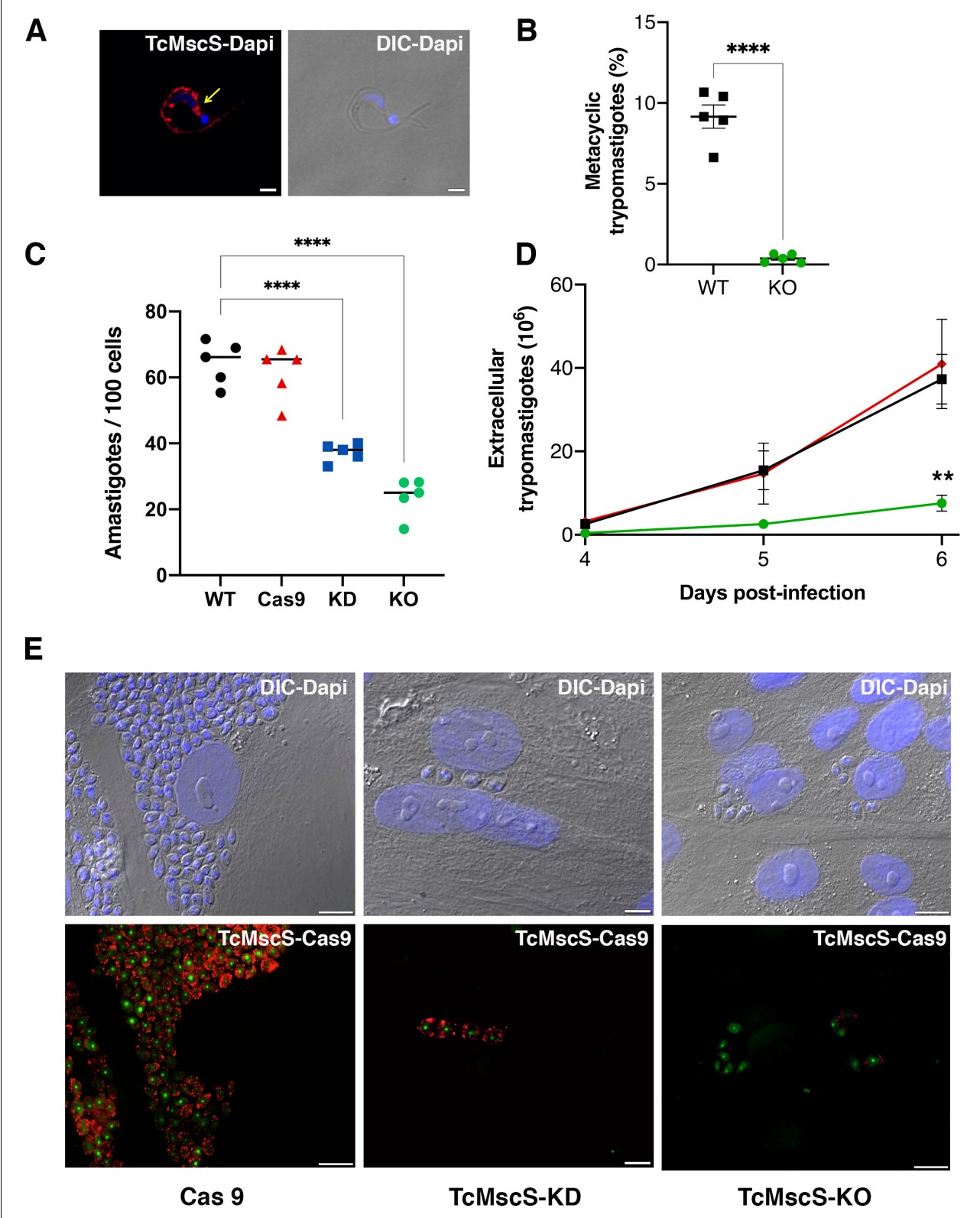

**Figure 8.** Infectivity defects in TcMscS mutants. (**A**) Representative immunofluorescence images showing the localization of TcMscS in metacyclic trypomastigotes labeled with anti-TcMscS antibodies (red). The position of the contractile vacuole is indicated by yellow arrows. Nuclei and kinetoplasts were 4′, 6-diamidino-2-phenylindole (DAPI) stained. Bar size = 2 µm. (**B**) Percentage of metacyclic forms obtained in vitro from wild-type Y strain (WT) and TcMscS-KO parasites incubated for 72 hr in TAU 3AAG media. Values are mean ± SEM of five independent experiments. (**C**) Quantification of

*Figure 8 continued on next page*

*Figure 8 continued*

intracellular amastigotes at 48 hr post-infection. HEK-293 cells were infected at a multiplicity of infection (MOI) of 25:1 with trypomastigotes WT (black), Cas9 (red), TcMscS-KD (blue), or TcMscS-KO (green). TcMscS mutants show a significant decrease in intracellular amastigotes. Values are mean ± SEM of five independent experiments in triplicate. (D) Quantification of extracellular trypomastigotes released from infected HEK-293 cells after 4 days of infection. Cells were infected at an MOI of 25:1 with WT (black), Cas9 (red), or TcMscS-KO (green) trypomastigotes and extracellular parasites were collected daily for counting. Values are mean ± SEM of four independent experiments. (E) Representative images of HEK-293 cells infected with Cas9 (Cas9), TcMscS-KD, or TcMscS-KO parasites. TcMscS was detected with specific antibodies (red); Cas9 was labeled with anti-GFP; nuclei and kinetoplasts were DAPI stained. Bar size = 10 μm. (**p<0.01, ****p< 0.0001).
The online version of this article includes the following source data for figure 8:

**Source data 1.** Quantification of infectivity data.

(*de Melo et al., 2008*; *Melo et al., 2006*; *Melo et al., 2004*; *Mott et al., 2009*). At this stage, TcMscS main role could be detecting changes in tension and shear stress elicited by host cell contraction. Alternatively, these results can be interpreted considering a simple topological factor: the spherical amastigotes would be more sensitive to osmotic downshock, whereas elongated forms have room for swelling and are better protected by a stronger cytoskeleton, explaining the change of TcMscS localization in amastigotes. This indicates the 'Laplacian disadvantage' of this stage that requires protective valves in the vulnerable unfolded outer membrane. The massive relocation of TcMscS points to compressive and shear stresses encountered in contractile mammalian tissues by non-motile forms with an altered subpellicular microtubular array (*Sinclair and de Graffenried, 2019*). In addition, the channel may be facilitating cell volume adjustments in the course of active cell reshaping during differentiation to trypomastigotes, which are preparing to exit the cells.

TcMscS expression is developmentally regulated, with higher levels in epimastigotes at the exponential phase of growth. Changes in expression and localization of the channel could be responding to stage-specific cues, as well as external conditions, allowing the parasites to successfully transition through their developmental cycle. To test this hypothesis, we used CRISPR/Cas9 (*Lander et al., 2015*) to target TcMscS in the parasites and produced two strains with similar characteristics but different levels of phenotypic defects. Neither the reduction (KD) nor the elimination (KO) of the gene resulted in a lethal phenotype, but both cell lines showed a detectable increase in cell volume, a significant reduction in their extracellular and intracellular replication rates, and impaired ability to produce metacyclic and bloodstream trypomastigotes, decreasing the infectivity of the parasites. The role of mechanical cues in differentiation has been demonstrated in several cell types including mesenchymal stem cells (*Corrigan et al., 2018*), osteoclasts (*Li et al., 2018*), and neurons (*Koser et al., 2016*) where TRPV and PIEZO 1 mechanosensitive channels' activity regulates developmental transitions (*He et al., 2019*; *Lu et al., 2016*) and axonal growth (*Koser et al., 2016*). Here, we provide the first evidence that mechanosensation mediated by TcMscS influences the developmental transition from non-infective to infective forms in *T. cruzi*, supporting the role of these channels in cell differentiation.

TcMscS-KO strains exhibited reduced motility and loss of infectivity. This is consistent with recent studies showing that motility is necessary for virulence in *T. brucei* (*Shimogawa et al., 2018*). In trypanosomatids, flagellar beating and motility are regulated by $Ca^{2+}$ transients generated in a directional manner along the flagellum (*Edwards et al., 2018*). The generation of $Ca^{2+}$ waves linked to the activation of mechanosensitive channels has been demonstrated in *Chlamydomonas* (*Collingridge et al., 2013*), tubular kidney cells (*Nauli et al., 2008*; *Wu et al., 2007*), lung epithelial lining (*Schmid and Salathe, 2011*), and bone cells (*Corrigan et al., 2018*). In *T. brucei*, a putative $Ca^{2+}$ channel has been shown to be essential for flagellar attachment and cell viability (*Shimogawa et al., 2015*), suggesting that $Ca^{2+}$ influx is required for normal function. In *T. cruzi*, an ortholog of this $Ca^{2+}$ channel has been identified in the CVC (*Ulrich et al., 2011*). The role of the CVC as an acidic calcium store has been documented in yeast (*Cunningham and Fink, 1994*), plants (*Martinoia et al., 2007*), and protists, including *Dictyostelium* (*Malchow et al., 2006*) and *T. cruzi* (*Rohloff and Docampo, 2008*; *Ulrich et al., 2011*). In these organisms, $Ca^{2+}$ ATPases mediate the accumulation of this ion in the CVC, while the efflux mechanisms vary. In *Dictyostelium*, $Ca^{2+}$ release is mediated by activation of a P2X receptor (*Ludlow et al., 2009*; *Parkinson et al., 2014*), while in *Paramecium*, this effect is caused by activation of IP3 receptors (*Ladenburger et al., 2006*). In *T. cruzi*, whether there is a specific channel that mediates $Ca^{2+}$ release from the CVC is still unknown.

Our results show a dysregulation in the $Ca^{2+}$ responses of TcMscS-KO, both at rest and after stimulation with calcium and Bay K. TcMscS seems to participate in the modulation of $Ca^{2+}$ dynamics either directly, by regulating calcium release from intracellular stores like the CVC (*Docampo et al., 2013*; *Docampo and Huang, 2015*), or indirectly by changing the activation of voltage-gated calcium channels associated with a non-canonical SOCE mechanism, similar to what has been observed in *T. gondii* (*Pace et al., 2014*). This association between the role of CVC in calcium regulation and TcMscS could also explain the observed defects in motility and infectivity (*Videos 2* and *3*).

In yeast and plants, MscS-like channels localized intracellularly regulate cytosolic $Ca^{2+}$ oscillations required for organelle homeostasis and volume regulation (*Lee et al., 2016*; *Nakayama et al., 2012*; *Nakayama et al., 2014*; *Nakayama et al., 2007*; *Wilson et al., 2011*; *Wilson et al., 2014*), supporting our hypothesis that TcMscS could be playing similar roles in *T. cruzi*.

As expected, the mutant parasites also showed a decreased ability to control cell volume both under stress and normal conditions, confirming the role of TcMscS in osmotic homeostasis. The compromised Regulatory Volume Decrease (RVD) observed in the knockout parasites suggests that other channels or transporters could be involved in the compensatory responses. This has been well documented in bacteria, where double knockouts of MscS and MscL are required to elicit a defect in osmoregulation. In *T. cruzi*, bioinformatic analysis shows the presence of genes encoding putative Piezo-like (*Prole and Taylor, 2013*) and TRP-like channels (*Cruz-Bustos et al., 2018*). Our RNA sequencing analysis of TcMscS-KO parasites does not show a significant change in the expression of those genes (data not shown). These results do not rule out the possibility of an increase in channel activity, independent of the mRNA expression level.

In summary, we have identified and functionally characterized the first mechanosensitive channel in trypanosomatids. TcMscS resembles bacterial MscS in its core, but lacks the first transmembrane helix and has a modified cytoplasmic domain. Beyond its canonical role in osmoregulation, the channel participates in calcium homeostasis, and its ablation affects the capacity of the cells to proliferate, move, differentiate, and infect host cells. Our work opens new avenues to explore the connection between sensing of physical stimuli and the activation of signaling pathways implicated in differentiation and host-parasite interaction. It also identifies new effectors that bridge sensing of external cues and the factors that drive stage transitions of protozoans in a host-specific manner. Additionally, these types of channels are not present in mammalian cells, offering potential as novel and selective targets for drug development.

## Materials and methods

**Key resources table**

| Reagent type (species) or resource | Designation | Source or reference | Identifiers | Additional information |
|---|---|---|---|---|
| Strain, strain background (*E. coli*) | TOP10 | Invitrogen | C404010 | Cloning bacteria |
| Strain, strain background (*E. coli*) | BL21 (DE3) pLysS | Invitrogen | C606010 | Expression bacteria |
| Strain, strain background (*E. coli*) | MJF641 (Δ7) | Provided by Dr. Ian Booth (University of Aberdeen, UK) | | Spheroplast strain |
| Cell line (*T. cruzi*) | Y strain | ATCC | 50832 | Parasites used in this study |
| Cell line (*Homo sapiens*) | HFF | ATCC | SCRC-1041 CVCL_3285 | Cells used to produce infective forms |
| Cell line (*H. sapiens*) | HEK293 | BEI resources | NR-9313 CVCL_0045 | Cells used to produce infective forms |

*Continued on next page*

*Continued*

| Reagent type (species) or resource | Designation | Source or reference | Identifiers | Additional information |
|---|---|---|---|---|
| Transfected construct | Human T Cell Nucleofector Solution | LONZA | VPA-1002 | Transfection reagent for epimastigotes |
| Recombinant DNA reagent | pQE80L | QIAGEN | 32943 | Bacterial expression vector |
| Recombinant DNA reagent | Cas9/pTREXn | Addgene | RRID:Addgene_68708 | Vector for CRISPR-Cas9 -mediated KO |
| Recombinant DNA reagent | tdTomato/ pTREXb | Addgene | RRID:Addgene_68709 | Vector for KO donor cassettes |
| Recombinant DNA reagent | pUC_sgRNA | Addgene | RRID:Addgene_68710 | sgRNA backbone containing vector |
| Recombinant DNA reagent | pTREXn-eGFP | Addgene | RRID:Addgene_62544 | Expression vector for *T. cruzi* |
| Antibody | Anti-TcMscS | Cocalico Biologicals Inc | Generated for this work | Polyclonal antibody produced in guinea pig used for western blot and IFA analysis (1:1000 and 1:100, respectively). |
| Antibody | Anti-myc (mouse monoclonal) | SIGMA | RRID:AB_439694 (M4439) | Antibody used for western blot analysis and IFA (1: 5000 and 1:250, respectively) |
| Antibody | Anti-actin (mouse monoclonal) | SIGMA | RRID:AB_262137 (A3853) | Antibody used for IFA (1:250) |
| Antibody | Anti-Tubulin (mouse monoclonal) | SIGMA | RRID:AB_477579 (T5168) | Antibody used for IFA (1:500) and western blot (1:5000) |
| Antibody | Anti-trypanosome SSP1 (mouse monoclonal) | BEI Resources | NR-50891 | Used as trypomastigote marker for IFA (1:250) |
| Antibody | Anti-trypanosome SSP4 (mouse monoclonal) | BEI Resources | NR- 50892 | Used as amastigote marker for IFA (1:250) |
| Antibody | Anti- GFP (rabbit polyclonal) | Thermo Fisher | RRID:AB_221570 (A6455) | Antibody used for IFA (1:3000) |
| Chemical compound, drug | Fura2-AM | Thermo Fisher | F1221 | Reagent used for calcium measurements |
| Software, algorithm | pCLAMP11 | Molecular Devices | RRID:SCR_011323 | Electrophysiology acquisition and analysis software |
| Software, algorithm | Graph pad PRISM9 | http://www.graphpad.com | RRID:SCR_002798 | Statistical analysis and graph software |

### TcMscS sequence analysis

Predicted protein sequences for TcMscS CL Brener Esmeraldo-like (TcCLB.504171.40) and Non-Esmeraldo-like haplotypes (TcCLB.509795.40) were compared with the putative sequences for *T. brucei* (Tb427.10.9030) and *L. major* (LmjF.36.5770) MscS-like channels (*Figure 1A*). Multiple alignments were compared with sequences for *E. coli* (WP_000389819) and *Helicobacter* pylori (WP_000343449.1) MscS channels. All the analyses were performed in Clustal Omega (https://www.ebi.ac.uk/Tools/msa/clustalo/). Linear topology predictions were done comparing the transmembrane domain predictions of TopPred, TMpred, and TMHMM 2.0 (https://www.expasy.org/tools/). Sequence similarity was analyzed using SIAS online tool (http://imed.med.ucm.es/Tools/sias.html). BLOSUM62, PAM250, and GONNET indexes were used to compare TcMscS and other MscS-like protein sequences.

## Bioinformatics and structural modeling

For modeling, the TcMscS sequence was independently submitted to six servers/predictors (I-TASSER, Robetta, Phyre2, SWISS-MODEL, IntFOLDTS, and RaptorX). The existing structures of *E. coli* MscS (2oau, 2vv5, 4agf, 4hwa, 5aji) and its homologs from *Thermoanaerobacter tengcongensis* (3t9n TtMscS, 3udc chimera of TtMscS, and EcMscS beta barrel) and from *H. pylori* (4hw9) served as templates. All acceptable monomeric predictions were assembled into heptamers using SymmDoc. After visual and automated inspection of the models, including identifying consensus trends and filtering out models that did not match the hydrophobicity profile of the membrane, we have selected one model (produced by IntFOLD2) as the most likely candidate for the closed state conformation of TcMscS. We resolved small structural conflicts and manually adjusted the positions of C-terminal amphipathic helices to form a coiled-coil bundle, similar to what was observed in the crystal structure of another type of bacterial mechanosensitive channel, MscL (2oar). The whole assembly was embedded into a pre-equilibrated POPC bilayer and refined in molecular dynamic (MD) simulations (CHARMM36 force field) using two cycles of unrestrained relaxation (5 ns) and symmetry-driven simulated annealing (1 ns).

The predicted open conformation of TcMscS was derived from the closed-state model of the channel using the EMP (*Anishkin et al., 2008a*; *Anishkin et al., 2008b*). This approach explores the conformational space of a protein by selecting and directionally propagating spontaneous thermal drift of the structure. It uses iterative cycles of extrapolated displacements (along the direction of motion taken by the system in the previous cycle), energy minimizations, and short MD simulations. Extrapolated simulations of multimeric axisymmetric channel complexes were run under symmetry constraints producing concerted motions. For TbMscS, the motion was initiated with a small thermal fluctuation of all atoms of the transmembrane barrel and the top of the cytoplasmic domain. A sequence of 50–100 extrapolation/relaxation cycles produced pseudo-continuous trajectories revealing substantial conformational changes while preserving most of the secondary structures. We have generated about 50 such trajectories, which explored a wide range of potential conformational transitions in TcMscS and produced 5400 expanded structures. The criteria for selection of candidate conformations from this pool were the lateral expansion of the transmembrane barrel, the diameter of the conductive pathway, and the retraction of the second gate residues to allow ion conductance. In addition, all transitions must produce minimal changes in the secondary structure of the channel. The analysis revealed a promising candidate for the open state characterized by a moderate displacement and tilting of TM3 helices that widened the main gate from 3.9 to 6.5 Å, which is associated with an in-plane area expansion of approximately 5.0 nm$^2$ (see *Figure 1* and *Video 1*). This expansion gave an estimation of 0.4 nS unitary conductance, which corresponds to the experimental value (*Figure 2*). The final extrapolated model of the open state was subjected to two cycles of refinement (5 ns unrestrained simulation followed by 1 ns symmetry annealing per cycle) in a fully hydrated POPC membrane with ions and was found stable under a tension of 20 dyne/cm for 10 ns. The structure of the C-terminal bundle and the position of the protein complex in the simulated bilayer are shown in *Figure 1—figure supplement 2*.

## Electrophysiological recordings of TcMscS

For electrophysiological studies in *E. coli* giant spheroplasts, we used an N-terminally 6-His-tagged version of *TcMscS* cloned into the expression plasmid pQE80 (Qiagen) with restriction sites BamHI and HindIII. TcMscS was expressed in the MJF641 (aka Δ7) strain devoid of seven endogenous MS channel genes (*Edwards et al., 2012*). The strain was a gift from Dr. Ian Booth (University of Aberdeen, UK). Competent MJF641 cells were transformed, selected overnight for transformants on plates with 100 µg/ml ampicillin, and used immediately for spheroplast preparation. Spheroplast preparation was done following standard protocols (*Delcour et al., 1989*). Briefly, 3 ml of the colony-derived culture was transferred into 27 ml of Luria-Bertani medium containing 0.06 mg/ml cephalexin to block septation. After 1.5–2 hr of incubation, 1 mM isopropyl β-D-1-thiogalactopyranoside (IPTG) was added to induce protein expression for 20 min. The filaments were transferred into a hypertonic buffer containing 1 M sucrose and subjected to digestion by lysozyme (0.2 mg/ml) in the presence of 5 mM ethylenediaminetetraacetic acid (EDTA) resulting in spheres of 3–7 µm diameter. The reaction was terminated by adding 20 mM Mg$^{2+}$. Spheroplasts

were separated from the rest of the reaction mixture by sedimentation through a one-step sucrose gradient.

## Recording conditions and protocols

Borosilicate glass (Drummond 2-000-100) pipettes with tips 1–1.3 µm in diameter (bubble number 4.9–5.1) were used to form tight seals with the inner membrane. The tension-activated channel activities were recorded in excised inside-out patches. In most experiments, the pipette and bath solution were symmetrical (200 mM KCl, 50 mM $MgCl_2$, 5 mM $CaCl_2$, 5 mM 4- (2-hydroxyethyl) -1-piperazineethanesulfonic acid (HEPES), pH 7.4). The bath solution was supplemented with 400 mM sucrose to osmotically balance the spheroplasts. An asymmetric configuration, 425 mM/200 mM KCl (bath/pipette), was used to determine the channel's anion/cation selectivity.

Calcium permeability was measured with a pipette solution containing 100 mM Ca gluconate, 20 mM $CaCl_2$, 5 mM HEPES, pH 7.2, and a bath solution containing only 100 mM Ca gluconate. The activities of TcMscS were recorded under a pressure ramp of 0 to −200 mmHg, at +30 mV or −30 mV voltage in the pipette. Traces were recorded using Clampex 10.3 software (MDS Analytical Technologies). Programmed mechanical stimuli were delivered using a modified high-speed pressure clamp apparatus (HSPC-1; ALA Scientific Instruments).

## Cell culture

WT epimastigotes were cultured in Liver Infusion Tryptose (LIT) medium as previously described (*Bone and Steinert, 1956*). Epimastigotes transfected with pTREX constructs, including Cas9scr and TcMscS mutants (TcMscS-KD and TcMscS-KO), were cultured in LIT media supplemented with 10% heat-inactivated fetal bovine serum (FBS-HI) and selection antibiotics (250 µg/ml of geneticin (G-418) and 15 µg/ml of blasticidin) (*Lander et al., 2015*).

Tissue-derived bloodstream trypomastigotes and amastigotes were obtained as previously described (*Moreno et al., 1994*). Human foreskin fibroblast (HFF- CVCL_3285) and HEK-293 (CVCL_0045) cells were cultured in High Glucose Dulbecco's Modified Eagles Media (HG-DMEM) supplemented with 100 units/ml of penicillin, 100 µg/ml of streptomycin, 0.2% amphotericin B, and 10% FBS-HI, and incubated at 37°C with 5% $CO_2$. The cell lines used in this work were obtained from ATCC and BEI Resources. Cell authentication and mycoplasma analysis were performed by the sources and documentation was provided at the time of purchase.

## In vitro metacyclogenesis

Differentiation to metacyclic trypomastigote forms was induced under chemically defined conditions using TAU medium as described (*Contreras et al., 1985*). Epimastigotes at 4 days of growth were collected by centrifugation at 1600 ×*g* for 10 min, washed once in phosphate-buffered saline solution (PBS) pH 7.4, resuspended in TAU media, and incubated 2 hr at 28°C. The supernatant was collected and resuspended in TAU with amino acids (TAU3AAG), incubated for up to 7 days at 28°C, and collected by centrifugation resuspended in 5 ml of Dulbecco's Modified Eagles Media (DMEM) supplemented with 20% fresh FBS to eliminate residual epimastigotes.

## In vitro infection assays

HEK-293 cells were plated onto coverslips in 12-well plates (1000 cells/well) and incubated in supplemented HG-DMEM overnight at 37°C with 5% $CO_2$. Infections were performed at a multiplicity of infection (MOI) of 25:1 with either WT, Cas9scr, or TcMscS mutant trypomastigotes. After 6 hr, the cells were washed three times with Hank's media and fresh DMEM was added. Coverslips were fixed in 4% paraformaldehyde-PBS at 6, 24, and 48 hr, stained with 4′,6-diamidino-2-phenylindole (DAPI) (5 µg/ml), and mounted in Fluoromont media for quantification of intracellular parasites or processed for immunofluorescence. IFAs were performed to verify expression of Cas9-GFP and TcMscS in intracellular amastigote forms. Cells were incubated with anti-TcMscS (1:100) antibody and anti-GFP rabbit serum (Thermo Fisher Scientific, Inc; RRID:AB_221570) (1:3000) antibodies. Pictures were acquired in an Olympus IX83 microscope and processed with CellSens Dimension. All infection quantifications were done in three coverslips per experiment, in five independent experiments. At least 100 host cells were quantified per coverslip. The number of host cells vs parasites was compared by one-way analysis of variance (ANOVA) with Dunnett's post-test.

## Quantification of extracellular trypomastigotes

The amount of infective forms released from HEK-293 cells infected at an MOI of 25:1 was evaluated by collecting the supernatant of the cultures at days 4, 5, and 6 post-infection. To eliminate possible contamination with detached cells, the supernatant was first centrifuged at 300 x$g$ for 5 min. The supernatant containing trypomastigotes was then centrifuged at 1600 x$g$ for 5 min and resuspended in 1 ml of DMEM. The cells were counted in a Z2 Beckman Coulter cell counter. All quantifications were done in triplicate for four independent experiments.

## TcMscS cloning, expression, and antibody generation

The complete ORF of *TcMscS* was amplified with primers 1 and 2 (*Supplementary file 1*-Table 1) from total genomic DNA of Y strain epimastigotes, cloned into pCR-Blunt II TOPO vector (Invitrogen), and sub-cloned into the pQE80 bacterial expression vector with restriction sites BamHI and HindIII. Bacterial expression of *TcMscS* ORF in BL21 plysS was induced overnight at 37°C with 1 mM IPTG. Recombinantly expressed TcMscS protein was extracted under denaturing conditions and purified with a nickel-agarose affinity column (Pierce) as previously reported (*Jimenez and Docampo, 2012*). Polyclonal antibodies against the whole protein were obtained by guinea pig immunization, following standard protocols (Cocalico Biologicals Inc, Reamstown, PA).

## Western blot analysis

For western blot analysis, parasites were collected at 1600 x$g$ for 10 min, washed twice in PBS, pH 7.4, and resuspended in modified radioimmunoprecipitation assay buffer (150 mM NaCl, 20 mM Tris-Cl, pH 7.5, 1 mM EDTA, 1% sodium dodecyl sulfate (SDS), and 0.1% Triton X-100) containing protease inhibitor cocktail (*Jimenez and Docampo, 2012*). Total homogenates were separated by sodium dodecyl sulphate–polyacrylamide gel electrophoresis, transferred onto nitrocellulose membranes, and blocked overnight with 5% nonfat dry milk in PBS-0.1% Tween 20 (PBS-T). Blotting was done with anti-TcMscS (1:1000) or anti-myc (1:250). Secondary antibodies conjugated with horseradish peroxidase were used when indicated. Membranes were stripped with 62.5 mM Tris-HCl, pH 6.8, 2% SDS, and 1% β-mercaptoethanol at 50°C for 30 min, washed in PBS-T, and incubated with monoclonal α-tubulin (Sigma) (1:5000, RRID:AB_477579) as a loading control.

## Immunofluorescence analysis

Epimastigotes, bloodstream trypomastigotes, and host cells containing intracellular amastigotes were fixed for 30 min in 4% paraformaldehyde. Fixed cells were attached to poly-L-lysine-treated glass coverslips for 10 min. Samples were permeabilized with 0.3% Triton X-100 for 3 min, washed in 1x PBS three times, and incubated in 50 mM NH$_4$Cl for 30 min at room temperature. After blocking overnight at 4°C in 3% bovine serum albumin solution, the cells were incubated with antibodies against TcMscS (1:500), actin (1:250, RRID:AB_262137), tubulin (1:500, RRID:AB_477579), SSP1 (1:250), or SSP4 (1:250) as indicated. Anti-SSP1 (# NR-50891) and anti-SSP4 (# NR050892) were obtained from BEI Resources, NIAID, NIH. The secondary antibodies were conjugated with Alexafluor 488 or 594 (1:3000) (Thermo Fischer Scientific, Inc, Waltham, MA). Coverslips were mounted with Fluoromount-G (SouthernBiotech, Birmingham, AL) containing DAPI (5 µg/ml). To evaluate the localization of TcMscS under osmotic stress, epimastigotes exposed to hypo- or hyperosmotic conditions were fixed at times 0, 2, and 5 min post-stress and processed for IFA as indicated before (*Jimenez and Docampo, 2012*). Immunofluorescence samples were imaged in an Olympus IX83 inverted microscope system and processed with CellSense Olympus software. Alternatively, samples were imaged in an ELYRA S1 (SR-SIM) Super Resolution Microscope (Zeiss) and processed with ZEN 2011 software, kindly provided by the Biomedical Microscopy Core at the University of Georgia.

## TcMscS mutant generation by CRISPR/Cas9

Two strategies were developed to generate TcMscS-KD and TcMscS-KO mutants using the CRISPR/Cas9 one-vector system (*Lander et al., 2015*). WT epimastigotes were transfected only with sgRNA/Cas9/pTREXn vector containing an sgRNA targeting *TcMscS*. Alternatively, the single vector containing TcMscS-sgRNA was co-transfected with a donor DNA to induce double-strand break repair by homologous recombination, carrying a blasticidin resistance marker and 100 bp flanking regions of *TcMscS*. Three protospacer adjacent motif (PAM) sites within the *TcMscS* gene were identified and

sgRNAs were designed to specifically target the gene. To avoid off-target effects, the protospacers were screened against the whole *T. cruzi* genome using ProtoMatch-V 1.0, kindly provided by Sebastian Lourido (*Sidik et al., 2014*). The sgRNAs targeting *TcMscS* were amplified with forward primers listed in *Supplementary file 1* (P7, P8, and P9) and reverse P10 using plasmid pUC_sgRNA as the template (*Lander et al., 2015*). The amplicons were cloned into TOPO-Blunt vector and subcloned into sgRNA/Cas9/pTREXn vector. Successful cloning of TcMscS-sgRNAs was confirmed by sequencing. Two donor templates were constructed using the gene encoding blasticidin-S deaminase (*Bsd*) amplified from tdTomato/pTREX-b (*Lander et al., 2015*) and the ultramers indicated in *Supplementary file 1* (P9–12). The first donor template contained 5'- and 3'-UTR regions of *TcMscS* flanking the *Bsd* cassette. The second donor template had the *Bsd* gene flanked by the last 100 bp of coding sequence of *TcMscS* and 100 bp of the 3'-UTR. Plasmids containing a scrambled sgRNA and Cas9 in a pTREXn vector were used as controls. To simplify the labeling, the cell line expressing Cas9 and the scrambled sgRNA is labeled Cas9.

## Cell transfections

Y-strain epimastigotes were grown for 4 days under standard conditions, collected at 1600 ×*g* for 10 min, washed once with buffer-A with glucose (BAG), and resuspended in 100 µl of Human T Cell Nucleofector Solution (Lonza). $5 \times 10^7$ cells were mixed with 5 µg of each sgRNA plasmid with or without 5 µg of DNA donor and electroporated under the U-033 Amaxa Nucleofector protocol. Cells were allowed to recover overnight and appropriate antibiotics for selection (250 µg/ml G-418 and 15 µg/ml blasticidin) were added.

## Genomic screening of transfectants

Clonal populations were obtained by serial dilutions, selected for 3 weeks, and a total of 12 clones were screened for proper gene targeting or replacement. Total genomic DNA was extracted from WT, Cas9, TcMscS-KD, and TcMscS-KO and analyzed by PCR to evaluate the presence of *TcMscS* (*Supplementary file 1*; P1, P2) or the DNA donors (*Supplementary file 1*; P3 and P4 for blasticidin, P5 and P6 for *TcMscS* downstream). GADPH was used as a housekeeping gene for loading control (*Supplementary file 1*; P15, P16). PCR results were confirmed by genomic DNA sequencing.

## Strain complementation

The complemented strains C1 and C2 (two independent clonal populations) were generated by introducing a copy of TcMscS in which the PAM sequence downstream of the sgRNA was mutated by PCR with primers P17 and P18 (*Supplementary file 1*-Table 1). The PAM-mutated ORF of *TcMscS* was amplified (primers P19, P20; *Supplementary file 1*) and subcloned into pTREX-p-myc with HindIII- XhoI restriction sites. The mutation was verified by sequencing and transfected into TcMscS-KO epimastigotes as described above.

Complemented strain C3 was obtained by overexpressing a copy of *T. brucei* homolog TbMscS. For that, Tb427.10.9030 sequence was amplified from genomic DNA extracted from WT procyclic forms of 427 strain (purchased from BEI Resources) with primers P21 and P22 (*Supplementary file 1*-Table 1), and subcloned into pTREX-p-myc with restriction sites HindIII-XhoI. The complementation plasmid was transfected into TcMscS-KO epimastigotes and the parasites were selected with puromycin (2.5 µg/ml) for 5 weeks. The expression of the myc-tagged proteins was verified by western blot with monoclonal anti-myc antibodies (1:5000, RRID:AB_439694).

## TcMscS expression levels

TcMscS mRNA expression was assessed by quantitative reverse transcription PCR (RT-qPCR) in the three life stages of the parasite as well as in WT, Cas9, TcMscS-KD, and TcMscS-KO. Epimastigotes were collected during the mid-log phase of growth, washed once with 1x PBS, pH 7.4, and homogenized in TRI Reagent. Total mRNA was extracted following the manufacturer's protocol (Sigma-Aldrich, St. Louis, MO) followed by chloroform/ethanol precipitation. cDNA was obtained using SuperScript III First-Strand Synthesis System (ThermoFisher Scientific, Inc, Waltham, MA) and oligo-dT (*O'Loughlin et al., 2013*) primers. cDNA was analyzed by quantitative PCR (qPCR) with Power SYBR Green PCR Master Mix (Applied Biosystems) and primers P1 and P2 for *TcMscS.* Trypomastigotes were collected by centrifugation from supernatants of infected HEK-293 cells at 5 days post-

infection. Amastigotes were collected by mechanical lysis of infected HEK-293 cells at 3 days post-infection. RNA extraction and cDNA synthesis were performed as indicated for epimastigotes. All qPCR results were normalized against GAPDH as a housekeeping gene (*Supplementary file 1*-Table 1; P15, P16) and indicated as $\Delta\Delta$Cq mean ± SD of at least three independent experiments in triplicate.

### Growth curves

Growth of epimastigotes was measured in TcMscS-KD, TcMscS-KO, WT, and Cas9 control parasites starting at a concentration of $1–5 \times 10^6$ cells/ml in LIT media supplemented as previously described. The number of parasites was determined daily using a Z Series Coulter Counter (Beckman Coulter). The results are expressed as mean ± SD of three independent experiments, in triplicate.

### Osmotic stress assays

Epimastigotes at the log phase of growth were collected at 1600 x*g* for 5 min, washed twice in PBS, and resuspended in isosmotic buffer (64 mM NaCl, 4 mM KCl, 1.8 mM $CaCl_2$, 0.53 mM $MgCl_2$, 5.5 mM glucose, 150 mM D-mannitol, 5 mM HEPES-Na, pH 7.4, 282 mOsmol/l) at a cell density of $1 \times 10^8$/ml. Aliquots of 100 µl of parasites were placed in a 96-well plate in isosmotic buffer and relative cell volume changes after osmotic stress were measured by light scattering method (*Jimenez and Docampo, 2012*). Briefly, pre-treatment readings of parasites in isosmotic buffer were taken for 3 min, followed by addition of 200 µl of hyposmotic buffer (64 mM NaCl, 4 mM KCl, 1.8 mM $CaCl_2$, 0.53 mM $MgCl_2$, 5.5 mM glucose, and 5 mM HEPES-Na, pH 7.4, were added for a final osmolarity of 115 mOsmol/l). Alternatively, the cells were subjected to hyperosmotic stress with 200 µl of hyperosmotic buffer (64 mM NaCl, 4 mM KCl, 1.8 mM $CaCl_2$, 0.53 mM $MgCl_2$, 5.5 mM glucose, 500 mM D-mannitol, 5 mM HEPES-Na, pH 7.4). The absorbance at 550 nm was measured every 10 s for 12 min. Readings were normalized against the average of 3 min of pre-treatment. Normalized absorbance readings were then converted into a percent volume change using the following equation: $[(V_F - V_i)/V_F] \times 100$, where $V_F$ is the absorbance value at the experimental time point and $V_i$ is the initial absorbance value obtained at the same time point under isosmotic conditions.

The maximum volume change was calculated by peak function analysis between 150 and 250 s. The final recovery volume was evaluated between 600 and 700 s. Results are the mean ± SE of six independent experiments in triplicate for all cell lines.

### Live video microscopy under osmotic stress

WT and TcMscS mutant epimastigotes were loosely adhered to poly-L-lysine-treated glass bottom coverslips under isosmotic conditions. After adherence, the epimastigotes were subjected to hyposmotic stress under the same conditions described above. Live video microscopy was taken under DIC illumination for 600 s in an Olympus IX83 inverted microscope system.

### Intracellular calcium measurements

Calcium measurements were done in WT (n = 12) and TcMscS-KO (n = 8) epimastigotes collected at 4 days of growth. Cells were pelleted at 1600 ×*g* for 10 min at room temperature, washed three times with BAG (in mM, NaCl 116, KCl 5.4, $MgSO_4$ 0.8, glucose 5, HEPES 50, pH 7.3) and loaded with 5 µM Fura2-AM (Molecular Probes) in BAG for 30 min at 30°C, washed twice, and resuspended in BAG at a concentration of $5 \times 10^8$ cells/ml. Aliquots of $5 \times 10^7$ cells were taken for each measurement with excitation at 340/380 nm and emission at 525 nm. Calibration was done by permeabilizing cells in BAG plus 1 mM ethylene glycol tetraacetic acid and then adding increasing concentrations of $CaCl_2$. The concentration of free calcium available was calculated using MaxChelator software (http://maxchelator.stanford.edu/) and the Kd was calculated according to the manufacturer's protocol. Experimental recordings were allowed to stabilize at baseline before addition of 1.8 mM $CaCl_2$ or 10 µM Bay K8644. Recordings were done on a Hitachi F7000 spectrofluorometer with excitation wavelength 340/380 nm and emission wavelength 510 nm.

### Statistical analysis

All the experiments were performed at least three times unless indicated otherwise, with intra-experiment replicates. The results are presented as the mean ± SD or SE as indicated in the figure

legends. Statistical significance was evaluated by one-way ANOVA with ad-hoc Bonferroni or Dunnett post-test or Student's t-test as indicated in the figures.

## Acknowledgements

We would like to thank Dr. Roberto Docampo for his guidance and manuscript revisions, and Stephen Vella and Dr. Myriam Hortua-Triana for assistance with super resolution imaging. We also thank Dr. Nikolas Nikolaidis for kindly providing antibodies against actin.

## Additional information

### Funding

| Funder | Grant reference number | Author |
| --- | --- | --- |
| National Institute of Allergy and Infectious Diseases | R15AI122153 | Veronica Jimenez |
| American Heart Association | 16GRNT30280014 | Veronica Jimenez |
| National Institutes of Health | 2T34GM008612-23 | Joshua Fonbuena |

The funders had no role in study design, data collection and interpretation, or the decision to submit the work for publication.

### Author contributions

Noopur Dave, Andriy Anishkin, Formal analysis, Investigation, Methodology, Writing - review and editing; Ugur Cetiner, Patricia Barrera, Formal analysis, Investigation, Methodology; Daniel Arroyo, Joshua Fonbuena, Megna Tiwari, Investigation, Methodology; Noelia Lander, Conceptualization, Methodology, Writing - review and editing; Sergei Sukharev, Conceptualization, Data curation, Formal analysis, Investigation, Visualization, Methodology, Writing - original draft, Writing - review and editing; Veronica Jimenez, Conceptualization, Data curation, Formal analysis, Supervision, Funding acquisition, Validation, Investigation, Visualization, Methodology, Writing - original draft, Project administration

### Author ORCIDs

Noopur Dave https://orcid.org/0000-0002-1798-1824
Ugur Cetiner https://orcid.org/0000-0002-5267-0749
Veronica Jimenez https://orcid.org/0000-0002-0744-8137

### Decision letter and Author response

Decision letter https://doi.org/10.7554/eLife.67449.sa1
Author response https://doi.org/10.7554/eLife.67449.sa2

## Additional files

### Supplementary files

• Supplementary file 1. Sequences of primers used. Bold letters indicate restriction sites. Protospacer sequences are underlined, and ultramer sequences that correspond to the flanking regions of the gene are in italics.

• Supplementary file 2. Changes in cell volume upon hypoosmotic stress. Peak and recovery analysis of epimastigotes' cell volume changes under hypoosmotic stress. For all the conditions, values are the mean ± SE of n = 6 independent experiments. p-values were calculated based on one-way analysis of variance with Bonferroni post-test. Differences were considered significant when $p < 0.01$(*).

• Supplementary file 3. Changes in cell volume upon hyperosmotic stress. Peak and recovery analysis of epimastigotes' cell volume changes under hyperosmotic stress. For all the conditions, values are the mean ± SE of n = 6 independent experiments. p-values were calculated based on one-way

analysis of variance with Bonferroni post-test. Differences were considered significant when p<0.01 (*).

• Supplementary file 4. Quantification of intracellular amastigotes. The number of intracellular amastigotes was counted at 6 and 48 hr post-infection. For all the conditions, values are the mean ± SE of n = 5 independent experiments. p-values were calculated based on one-way analysis of variance with Bonferroni post-test. Differences were considered significant when p<0.05(*).

• Supplementary file 5. Quantification of extracellular trypomastigotes. Extracellular trypomastigotes collected from the supernatant of infected cells were counted at days 4, 5, and 6 post-infection. For all the conditions, values are the mean ± SE of n = 4 independent experiments. p-values were calculated based on one-way analysis of variance with Bonferroni post-test. Differences were considered significant when p<0.05(*).

• Transparent reporting form

## Data availability

All data generated in this study are included in the manuscript, supporting files and source data files.

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
