## [Decision Letter]

**Acceptance summary:**

Mechanosensitive channels allow passage of small molecules in response to various mechanical stresses. *Trypanosoma cruzi* is an important human pathogen, which lives intracellularly within mammalian hosts and extracellularly in the digestive system of reduviid bugs. In this paper the authors show that a *Trypanosoma cruzi* mechanosensitive channel is important for osmoregulation and infectivity; interestingly, it is located on the contractile vacuole in the extracellular parasites, but on the plasma membrane of the intracellular form.

**Decision letter after peer review:**

Thank you for submitting your article "A novel mechanosensitive channel controls osmoregulation, differentiation and infectivity in *Trypanosoma cruzi*" for consideration by *eLife*. Your article has been reviewed by 3 peer reviewers, and the evaluation has been overseen by a Reviewing Editor and Dominique Soldati-Favre as the Senior Editor. The following individuals involved in review of your submission have agreed to reveal their identity: Kenjiro Yoshimura (Reviewer #1); Hannah Malcolm (Reviewer #2).

Essential revisions:

Please see the reviews for detailed suggestions. After discussion the reviewers have agreed that no further experiments are needed but some discrepancies and things that need further investigation should be highlighted in the text.

1) There is a notable difference between the mobility of recombinant TcMscS employed in the electrophysiological characterization and that of the protein expressed in parasites. This suggests potential proteolytic processing in parasites, and it would be interesting to determine what segment may be removed. The authors should highlight this difference in the text and provide the caveat that some properties of the recombinant channel could be altered in the context of the parasite.

2) Expand the mechanosensation section to give more details as proposed by Reviewer 2.

3) Mention that more detailed quantitative studies would be needed to clarify possible differences in motility.

*Reviewer #1 (Recommendations for the authors):*

This study provides comprehensive data regarding the structure and function of the MscS-like protein in *T. cruzi*. This paper is of high importance but there are several concerns to be addressed.

(1) The molecular mass of recombinant TcMscS is clearly larger than that of native TcMscS (Figure 3E) suggesting that TcMscS may be subjected to splicing or cleavage in *T. cruzi* cells. Has this possibility been tested? Considering that the recombinant TcMscS is obtained from *E. coli*, the characteristics of the channels assessed by electrophysiology should derive from the recombinant TcMscS, which may be different from native TcMscS.

(2) The authors state that motility was markedly reduced in the KO strain of TcMscS, but the data are presented exclusively in video. The finding that motility was impaired is interesting but should be described more thoroughly.

Aren't there any quantitative ways to describe the motility? Flagellar beat frequency could be measured. The cells probably twitched and did not swim because the flagella were attached to the poly-L lysine coated glass. How did the cells swim when they were not attached to glass surface? Can swimming velocity be measured?

It is not clarified why the motility was impaired. If the impairment is due to the lack of Ca^2+^ flow through TcMscS, WT may also show such reduction in motility in Ca^2+^-free solution. The motility impairment in KO strain may be rescued by addition of calcium ionophore.

(3) The TcMscS KO strain showed exaggerated responses to hypo- and hyper osmotic stress. Data supporting a link to the function CV should ideally be included. In other protists, the interval of CV contraction changes with osmotic stress (Kosmic-Buchmann et al., Enkaryotic Cell, 13: 1432-1430, 2014). Is live imaging of CV contraction possible in *T. cruzi*? It would be exciting if the CV of the TcMscS KO strain responds differently from the WT.

*Reviewer #2 (Recommendations for the authors):*

In the introduction and Figure 1 when the structures of EcMscS and the models of TcMscS are discussed the location of the lipids in Figure 1B are not consistent with the most recent EcMscS structures (PMID: 31880537 and PMID: 31291591).

*Reviewer #3 (Recommendations for the authors):*

1. In lines 153-155, the authors refer to single channel traces in Figure 2A as being recorded under 110 mmHg pressure steps. Where during the experiment were the pressure steps applied, or were these traces recorded without pressure jumps? The authors should indicate in the figure or the legend how these steps gated the channel, if this is the case.

2. Po and Pc should be defined in the legend to Figure 2F.

3. In lines 231, 233, and 236, the authors should indicate that it is the Figure 5 supplement that contains the relevant images.

4. The authors indicate that trypomastigotes exhibit staining for the TcMscS channel that is neither in the CVC nor the surface membrane but is interior to the cytosol and flagellum. Do the authors think this fluorescence represents the channel in intracellular vesicles? It may be worth commenting on this point.

5. At the bottom of pg. 12, the authors suggest that TcMscS may be involved in regulating intracellular stores of Ca^2+^ such as those in the CVC. It is puzzling that the knockout line does not exhibit the large increases in intracellular Ca^2+^ mediated by opening of VGCC channels in the plasma membrane by Bay K8644. However, I am not sure how regulating intracellular Ca^2+^ stores would provide the pronounced differences seen between wild type and knockout cells. This section needs clarification.

6. Line 703, legend to Figure 3. The statement should read "∆∆Cq with respect to…".

7.In the Western blot in Figure 5, supplement 2, there is a clear residual band at the position of TcMscS in the KO line. Could the authors comment on this fact, given that the mRNA appears to be absent in part C?

---

## [Author Response]

Reviewer #1 (Recommendations for the authors):This study provides comprehensive data regarding the structure and function of the MscS-like protein in T. cruzi. This paper is of high importance but there are several concerns to be addressed.(1) The molecular mass of recombinant TcMscS is clearly larger than that of native TcMscS (Figure 3E) suggesting that TcMscS may be subjected to splicing or cleavage in T. cruzi cells. Has this possibility been tested? Considering that the recombinant TcMscS is obtained from E. coli, the characteristics of the channels assessed by electrophysiology should derive from the recombinant TcMscS, which may be different from native TcMscS.

Trypanosomatids, in general, do not contain introns in their genome but instead have polycistronic units that are processed by trans-splicing and addition of splice leader sequences (Eukaryotic Cell, Aug. 2010, p. 1159–1170). Only a handful of genes containing introns have been described in *T. brucei*, but none in *T. cruzi*, and canonical cis- splicing of genes to generate different protein isoforms has not been documented in the parasites. TcMscS is annotated as a single exon gene, thus, it is unlikely that it is spliced to generate a shorter form of the protein. The recombinant protein in figure 3A is expressed in pQE80 vector and contains an N-terminal RGS-6xHis tag that adds about 2kDa to the native proteins observed in the parasites. While we cannot rule out the possibility of proteolytic processing of the native TcMscS, we believe the difference in motility could be due to the presence of the tag in the recombinant protein. We agree that there could be differences in the behavior of the native TcMscS compared with the one expressed in *E. coli*, but this does not invalidate the data obtained in spheroplasts, as this system has shown to reproduce accurately the mechanosensitive characteristics of other MscS channels.

(2) The authors state that motility was markedly reduced in the KO strain of TcMscS, but the data are presented exclusively in video. The finding that motility was impaired is interesting but should be described more thoroughly.Aren't there any quantitative ways to describe the motility? Flagellar beat frequency could be measured. The cells probably twitched and did not swim because the flagella were attached to the poly-L lysine coated glass. How did the cells swim when they were not attached to glass surface? Can swimming velocity be measured?It is not clarified why the motility was impaired. If the impairment is due to the lack of Ca^2+^ flow through TcMscS, WT may also show such reduction in motility in Ca^2+^-free solution. The motility impairment in KO strain may be rescued by addition of calcium ionophore.

We agree with the reviewer that the motility defects in the KO strains deserve further exploration. We have conducted a series of experiments to evaluate the phenotype, including the quantification of the motility by differential dynamic microscopy

(doi: 10.1016/j.bpj.2012.08.045), and observed a defect both in the ballistic (directional), as well as the diffusion coefficient (representing the random twirling that is characteristic of *T. cruzi* epimastigotes movement). We have also observed a defect in the formation of new flagella, with KO parasites exhibiting a significantly shorter flagella, and a decrease in the expression of flagellar proteins (at RNA and protein level) that could be related with abnormal trafficking of new components to assembly and maintain normal flagellar structures. This agrees with previous reports of protein trafficking through the contractile vacuole in *T. cruzi*, as, by nature the CVC is considered as a post-Golgi compartment (https://doi.org/10.1371/journal.ppat.1004224). The extensive data supporting the analysis of the motility phenotype will be published shortly in a separate manuscript. Author response image 1 is a representative figure of the motility analysis to be included in the future publication as well as SEM images of the KO and WT parasites where the shorter flagella can be observed (Author response image 2). Given the relevance and magnitude of the new collected data, we decided not to include it in the current manuscript as it stands by itself as the continuation of this work.

**Author response image 1. sa2fig1:** Motility analysis of WT and TcMscS KO by differential dynamic microscopy (DDM). Dp corresponds to diffusion coefficient and V the linear velocity. There is a significant difference in the speed of KO (0.7 μm/s) vs WT parasites (2.5 μm/s) as well as the diffusional movement (0.4 vs 2.5, respectively).

**Author response image 2. sa2fig2:** Morphological analysis of *T. cruzi* epimastigotes by Scanning Electron Microscopy. WT (A) and control epimastigotes expressing scrambled Cas 9 (B) show normal morphology and elongated flagella. TcMscS KO parasites (C-F) present short flagella and incomplete cytokinesis.

We use poly-lysine for imaging purposes because it slightly decreases the ballistic movement of the parasites, facilitating its visualization, but it does not critically change the robust movement of the cells. The differential dynamics microscopy analysis described above was done in absence of poly-lysine to avoid interference of external factors.

(3) The TcMscS KO strain showed exaggerated responses to hypo- and hyper osmotic stress. Data supporting a link to the function CV should ideally be included. In other protists, the interval of CV contraction changes with osmotic stress (Kosmic-Buchmann et al., Enkaryotic Cell, 13: 1432-1430, 2014). Is live imaging of CV contraction possible in T. cruzi? It would be exciting if the CV of the TcMscS KO strain responds differently from the WT.

We thank the reviewer for the suggestion of measuring the frequency of CV contraction. In the past, we have been successful at decreasing the motility of the parasites to obtain video microscopy of the CV but the twirling movement of the parasites usually precludes us from long term acquisition required for estimation of the frequency of CV contraction. The role of the CVC and acidocalcisomes as the main organelles responsible for cell volume regulation in *Trypanosoma cruzi* has been extensively documented by Dr. Roberto Docampo’s lab (Exp Parasitol. 2008 Jan;118(1):17-24). His group has shown that impairing the function of the CV severely decreases RVD responses (J Biol Chem. 2004 Dec 10;279(50):52270-81). While the direct link between the function of TcMscS and the function of the CV has not been established, the consistent phenotype of KD and KOs strongly suggest the role of channel in cell volume homeostasis, via the CV function. Definitive proof of this link could be provided by reversion of the phenotype in complemented strains, but as it was mentioned in the manuscript (lines 261264) the complementation had detrimental effects on the parasites (Figure 5 supplement 3) most probably due to overexpression toxicity.

Reviewer #2 (Recommendations for the authors):In the introduction and Figure 1 when the structures of EcMscS and the models of TcMscS are discussed the location of the lipids in Figure 1B are not consistent with the most recent EcMscS structures (PMID: 31880537 and PMID: 31291591).

The position of the transmembrane barrel in the lipid bilayer is shown in the new supplement to Figure 1. We cannot expect any consistency with the recent cryoEM data obtained on EcMscS because of the two proteins have different length and topology. As MD simulations show, the length of the TcMscS barrel is sufficient to cross the thickness of a typical POPC bilayer. Please see legend to the supplement to Figure 1.

Reviewer #3 (Recommendations for the authors):1. In lines 153-155, the authors refer to single channel traces in Figure 2A as being recorded under 110 mmHg pressure steps. Where during the experiment were the pressure steps applied, or were these traces recorded without pressure jumps? The authors should indicate in the figure or the legend how these steps gated the channel, if this is the case.

the legend to Figure 2 has been reworked.

2. Po and Pc should be defined in the legend to Figure 2F.

Po and Pc are now defined in page and in the legend to Figure 2.

3. In lines 231, 233, and 236, the authors should indicate that it is the Figure 5 supplement that contains the relevant images.

This has been corrected.

4. The authors indicate that trypomastigotes exhibit staining for the TcMscS channel that is neither in the CVC nor the surface membrane but is interior to the cytosol and flagellum. Do the authors think this fluorescence represents the channel in intracellular vesicles? It may be worth commenting on this point.

Additional comments and references have been incorporated in lines 214-216 and 398-400.

5. At the bottom of pg. 12, the authors suggest that TcMscS may be involved in regulating intracellular stores of Ca^2+^ such as those in the CVC. It is puzzling that the knockout line does not exhibit the large increases in intracellular Ca^2+^ mediated by opening of VGCC channels in the plasma membrane by Bay K8644. However, I am not sure how regulating intracellular Ca^2+^ stores would provide the pronounced differences seen between wild type and knockout cells. This section needs clarification.

While the role of acidocalcisomes and the CVC in intracellular calcium homeostasis have been previously documented, how the intracellular store management interacts with possible permeation pathways in the plasma membrane is unclear. We have expanded the discussion on the effect of TcMscS-KO in calcium homeostasis in page 19 and incorporated a few relevant references.

6. Line 703, legend to Figure 3. The statement should read "∆∆Cq with respect to…".

This has been corrected.

7. In the Western blot in Figure 5, supplement 2, there is a clear residual band at the position of TcMscS in the KO line. Could the authors comment on this fact, given that the mRNA appears to be absent in part C?

It is possible that the faint band in the KO is the result of a spillover. For clarity, we have replaced the western blot in Figure 5 Supplement 2A.